# Common Task Framework For a Critical Evaluation of Scientific Machine Learning Algorithms

Philippe M. Wyder[1],     Judah Goldfeder[2],     Alexey Yermakov[1,3],     Yue Zhao[4],

Stefano Riva[6],     Jan Williams[5],     David Zoro[3],     Amy Sara Rude[1],

Matteo Tomasetto[7],     Joe Germany[8],     Joseph Bakarji[9],     Georg Maierhofer[10],

Miles Cranmer[10],                    J. Nathan Kutz[1,3] *

[1]Department of Applied Mathematics, University of Washington, Seattle, WA 98195
[2]Department of Computer Science, Columbia University, New York, NY 10027
[3]Department of Electrical and Computer Engineering, University of Washington, Seattle, WA 98195
[4]High Performance Machine Learning, SURF, Amsterdam, the Netherlands
[5]Department of Mechanical Engineering, University of Washington, Seattle, WA 98195
[6]Department of Energy, Nuclear Engineering Division, Politecnico di Milano, Milan, Italy
[7]Department of Mechanical Engineering, Politecnico di Milano, Milan, Italy
[8]Department of Mathematics, American University in Beirut, Beirut, Lebanon
[9]Department of Mechanical Engineering, American University in Beirut, Beirut, Lebanon
[10]Department of Applied Mathematics and Theoretical Physics, University of Cambridge, Cambridge, UK

## Abstract

Machine learning (ML) is transforming modeling and control in the physical, engineering, and biological sciences. However, rapid development has outpaced the creation of standardized, objective benchmarks—leading to weak baselines, reporting bias, and inconsistent evaluations across methods. This undermines reproducibility, misguides resource allocation, and obscures scientific progress. To address this, we develop a Common Task Framework (CTF) for scientific machine learning. The CTF features a curated set of datasets and task-specific metrics spanning forecasting, state reconstruction, and generalization under realistic constraints, including noise and limited data. Inspired by the success of CTFs in fields like natural language processing and computer vision, our framework provides a structured, rigorous foundation for head-to-head evaluation of diverse algorithms. As a first step, we benchmark methods on two canonical nonlinear systems: Kuramoto-Sivashinsky and Lorenz. These results illustrate the utility of the CTF in revealing method strengths, limitations, and suitability for specific classes of problems and diverse objectives. Next, we are launching a competition based on a global, real-world sea surface temperature dataset with a true holdout dataset to foster community engagement. Our long-term vision is to replace ad hoc comparisons with standardized evaluations on hidden test sets, thereby raising the bar for rigor and reproducibility in scientific ML.

---

*Corresponding author: `kutz@uw.edu`

39th Conference on Neural Information Processing Systems (NeurIPS 2025) Track on Datasets and Benchmarks.

# 1 Introduction

Data science, especially machine learning (ML) and artificial intelligence (AI), is transforming almost every aspect of the engineering, physical, social, and biological sciences. As the body of literature on new ways to model many scientific data and systems grows, we still lack objective measures to adequately characterize and compare these methods. In the absence of a common standard for benchmarking new and existing approaches, the current literature suffers from weak baselines, reporting bias, and inconsistent evaluations [23], thus leading to a clear call for a CTF for evaluating machine learning methods on a diverse set of metrics related to science and engineering [17]. Several benchmark frameworks have been proposed to address this gap in scientific machine learning. For example, The Well [25] provides a large-scale collection of diverse physics simulation datasets across multiple domains. CoDBench [4] offers a comprehensive benchmarking suite to systematically evaluate data-driven models for solving differential equations and continuous dynamical systems. PDEBench [31] and PDEArena [12] are PDE-focused benchmarking frameworks that provide curated datasets and task suites to assess the accuracy and efficiency of ML-based solvers. These benchmarks exemplify the move toward standardized, reproducible evaluation in scientific ML. Nevertheless, despite the rise of benchmark data sets across science and engineering, the reliance on self-reporting has generated a significant reproducibility crisis. Self-reporting is, in general, a flawed premise. For instance, neural networks upon training are typically initialized with random weight assignment. Although the errors achieved on the training data set are comparable from run to run, the errors on the test set can be significantly different. This can lead to $p$-hacking, or judicious picking of results, when reporting scores on test data sets, i.e. simply re-train the model until a desired and good result is achieved for self-reporting. Only with a true, withheld test set is a comparison among methods possible.

CTFs play a critical role in evaluating methodological advancements. Donoho [9] has argued that the successful application of CTFs is a primary factor for the success of data science and machine learning. Indeed, the fields of speech recognition, natural language processing, and computer vision have developed mature CTF platforms that are progressively updated with more challenging data in order to drive progress and innovation. For instance, the industry-leading CVPR conference offers more than 30 challenge problems per year for participants to score and benchmark their ML/AI algorithms against. More broadly, classic fields of machine learning have benefited from extensive benchmark environments and common task frameworks, including ImageNet [8, 15], Go and chess [30], video games such as Atari [24] and StarCraft [33], the OpenAI Gym [29, 10], among other environments for more realistic control [7, 32]. Unlike these leading fields, many scientific disciplines have yet to integrate the CTF into their core infrastructure [23]. This compromises true comparative metrics between methods, algorithms, and results, and it limits the potential of ML in these areas.

## 1.1 Common Task Framework for Science and Enginering

We propose a CTF for science and engineering that is primarily focused on evaluating machine learning and AI models for dynamic systems: systems whose underlying evolution is determined by physical or biophysical principles or governing equations. The CTF will provide training data sets with clear and concise goals related to forecasting and reconstruction under various challenging scenarios, such as noisy measurements, limited data, or varying system parameters, as advocated in [17]. Given a training dataset and a range of timesteps to predict, users will produce approximations for a hidden test dataset. The predictions are evaluated and scored on a diverse set of metrics by an independent referee and posted on a leaderboard.

Scoring is by nature reductive—reducing a method's performance to a single floating point value. We choose a multi-metric scoring approach because a single number often doesn't provide enough information on whether a method is right for an application or not. As a result, we decided to carefully design a twelve-score system designed to match crucial tasks required in science and engineering. A summary, or composite score, is also produced that gives the overall score for a given method. Rankings by task and overall performance are highlighted here and tracked on a leader board.

To visualize the overall performance of a method, a radar plot is generated highlighting the various scores associated with the challenge (see Fig. 1). From this figure one can glean the characterization of a method with respect to its performance on the diverse set of CTF tasks. The average of all scores serves as the composite score. This scoring system prevents a winner takes all framework,

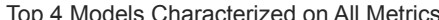

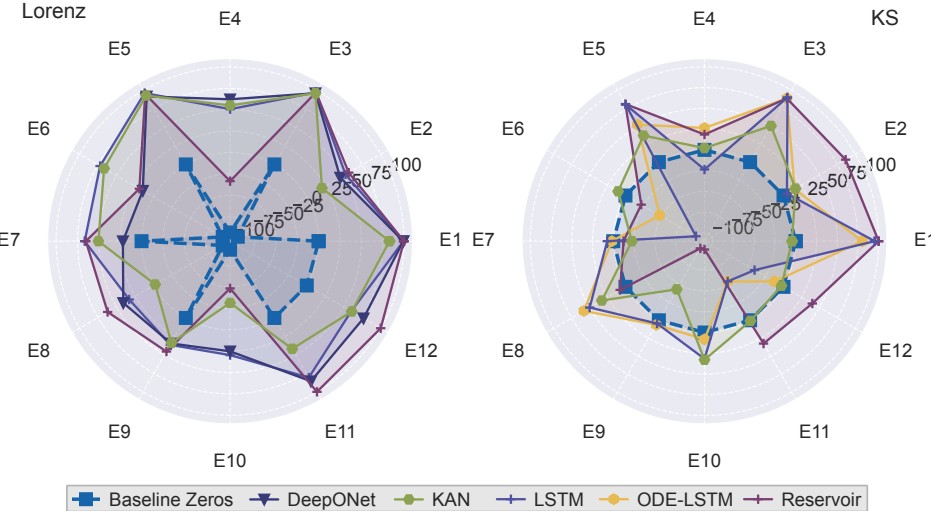

Figure 1: The twelve-axis radar plot characterizes a method's performance across all tasks on a dataset, and provides a visual performance profile. The axes correspond to the various tasks associated with forecasting and reconstruction with noise, limited data and parametric dependency. The chart shows the top four performing metrics on the KS and the Lorenz dataset scored against their reference baselines: constant zero and average prediction respectively.

since different modeling approaches will excel on different tasks. Some will do well with noise, others will not. Others might excel in the limited data regime, while performing poorly under parametric generalization. These profiles are important to provide a comprehensive and well-rounded performance metric, and help guide for scientists for selecting a suitable method.

Once the **ctf4science** is launched[2], we invite everyone to benchmark their methods on the CTF for Science by taking the following steps:

1. Sign-up and Sign-in on Kaggle
2. Train your model with our training data and generate predictions for each benchmark case
3. Submit prediction files to the competition platform
4. See your score on the leaderboard

To interact with **ctf4science** before the competition launch visit our GitHub repository[3], install the **ctf4science** package[35], and evaluate your method on our datasets *ODE_Lorenz*, *PDE_KS*, and *SST*. Our datasets and our **ctf4science** Python package don't require high-performance hardware and can be run on a laptop computer.

## 2   Datasets & Evaluation Metrics

We launch the CTF platform with two canonical and commonly used models in scientific machine learning: the Lorenz equations, a dynamical system and the Kuramoto-Sivashinsky (KS) equation, a partial differential equation. Both exhibit complex and challenging behavior for the science and engineering tasks of reconstruction and forecasting under the constraints of noise, limited data, and parametric dependence. While these equations serve as a starting point, the CTF will evolve to include both more complex data and more challenging tasks. The CTF framework is a sustainable platform that evolves and grows as the community develops more sophisticated methods and algorithms and faces new challenges.

---

[2]Kaggle launch date TBD

[3]Available at `https://github.com/CTF-for-Science/ctf4science`

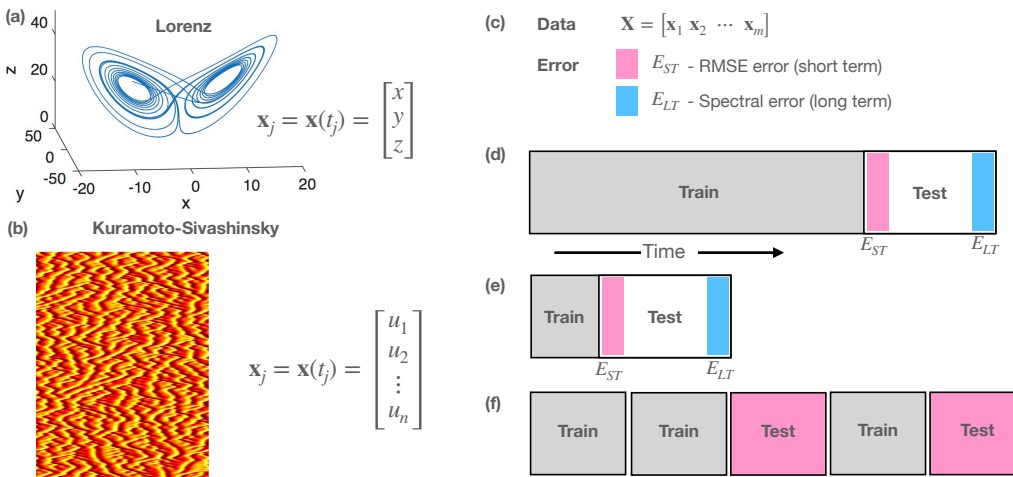

Figure 2: The CTF Evaluation framework scores the performance of methods on (a) the Lorenz dynamical system and (b) the Kuramoto-Sivashinsky partial differential equation. (c) Data is collected and organized into matrices which is then split into testing and training sets. RMSE errors are computed for reconstruction and short-time forecasting, while the spectral error computes the statistics of long-time forecasting (spatial or temporal). (d) Forecasting and reconstruction tasks are evaluated on noise-free, low-noise and high-noise data. Methods are also evaluated when (e) only limited data is available and (f) for reconstruction of parametrically dependent data.

We provide a detailed breakdown of the evaluation metrics and the associated data matrices in the following sections. For convenience, we included an overview table that summarizes the relationship between each evaluation metric and the corresponding data matrices in the supplementary materials.

## 2.1 Spatio-Temporal System: Kuramoto-Sivashinsky

The KS equation is a fourth order, nonlinear partial differential equation. It is considered a canonical example of spatio-temporal chaos in a one-dimensional PDE and is therefore commonly used as a test problem for data-driven algorithms. The KS equation is a particularly challenging case for fitting algorithms due to its combination of high dimensionality, nonlinearity, and sensitivity to initial conditions (chaotic behavior):

$$u_t + uu_x + u_{xx} + \mu u_{xxxx} = 0. \tag{1}$$

The solutions of Eq. (1) are defined on a grid across the domain of $[0, 32\pi]$ with periodic boundary conditions. A numerical integrator with an unknown time step $\Delta t$ evolves the solution $m$ steps.

### 2.1.1 Test 1: Forecasting (2 scores)

The first test of the method, as illustrated in Fig. 2-d, involves the approximation of the future state of the system. Thus, given a data matrix representing the dynamics over $t \in [0, 10T]$ ($\mathbf{X}_1 \in \mathbb{R}^{10m \times n}$), the forecast is requested for $t \in [10T, 11T]$ ($\mathbf{X}_{1\text{pred}} \in \mathbb{R}^{m \times n}$), with $n$ being the dimension of the system and $m$ being the number of time steps. The forecasting score is composed of two scores evaluating both the short-time forecast $E_{\text{ST}}$ (the "weather forecast"), which is computed using root-mean square error (RMSE) between the test set and the user's approximation, and the long-term forecast $E_{\text{LT}}$ (the "climate forecast"), which is based upon the power spectral density - see Fig. 2-c. As such, the following two error scores are computed:

$$S_{\text{ST}}(\tilde{\mathbf{X}}, \hat{\mathbf{X}}) = \frac{\|\hat{\mathbf{X}}_1[1:k,:] - \tilde{\mathbf{X}}_1[1:k,:]\|}{\|\hat{\mathbf{X}}[1:k,:]\|} \qquad \text{(weather forecast)} \tag{2}$$

$$S_{\text{LT}}(\tilde{\mathbf{X}}, \hat{\mathbf{X}}) = \frac{\|\hat{\mathbf{P}}[N-k:N,\mathbf{k}] - \tilde{\mathbf{P}}[N-k:N,\mathbf{k}]\|}{\|\hat{\mathbf{P}}[N-k:N,\mathbf{k}]\|} \qquad \text{(climate forecast)}. \tag{3}$$

For the challenge dynamics of interest, sensitivity of initial conditions is common, making long range forecasting to match the test set an unreasonable task given fundamental mathematical limitations

with Lyapunov times. Thus, as shown above, the long-time error is computed by least-squares fitting of the power spectrum $\mathbf{P}[k, :] = \ln(|\text{FFT}(\mathbf{X}[k, :])|^2)$, where the **fftshift** has been used to model the data in the wavenumber domain and $\mathbf{k} = n/2 - k_{max} : n/2 + (k_{max} + 1)$ with $k_{max} = 100$. This means that we look at the match in the first 100 wavenumbers of the power spectrum over a long time simulation. It is clear that there are many ways to evaluate the long-range forecasting capabilities. We chose a simple and transparent metric fully understanding that more nuanced scoring could be used. To provide a reasonable range we then compute the two scores

$$E_1 = 100(1 - S_{\text{ST}}(\mathbf{X}_{1\text{pred}}, \mathbf{X}_{1\text{test}})), \quad E_2 = 100(1 - S_{\text{LT}}(\mathbf{X}_{1\text{pred}}, \mathbf{X}_{1\text{test}})), \tag{4}$$

meaning in each case a score of $E_i = 100$ corresponds to a perfect match. Note that, as a baseline, a solution guess of zeros $\tilde{\mathbf{X}}_{1\text{pred}}[1 : k, :] = \mathbf{0}$ (corresponding also to $\tilde{\mathbf{P}}_{1\text{pred}}[N - k : N, \mathbf{k}] = \mathbf{0}$) gives a score of $E_1 = E_2 = 0$.

**Input:** $\mathbf{X}_{1\text{train}} \in \mathbb{R}^{10m \times n}$;  **Output:** $\mathbf{X}_{1\text{pred}} \in \mathbb{R}^{m \times n}$;  **Scores:** $E_1, E_2$.

### 2.1.2 Test 2: Noisy Data (4 scores)

The ability to handle noise is critical in all data-driven applications as sensors and measurement technologies are by default embedded with varying levels of noise. Methods that work with numerically accurate data, for example data points that are $10^{-6}$ accurate, may be useful for model reduction, but are rarely suitable for discovery and engineering design from real-world data. Both strong and weak noise are considered as these represent realistic challenges to be addressed in practice.

This test is very similar to Test 1, but now with noise added to the data. Specifically, the challenger is given a data matrix $\mathbf{X}_{2\text{train}} \in \mathbb{R}^{10m \times n}$ and $\mathbf{X}_{3\text{train}} \in \mathbb{R}^{10m \times n}$ representing the evolution with medium or high noise respectively. The objective is to first produce a reconstruction of the data itself, i.e. denoise the data to produce an estimate of the true state of the dynamics, $\mathbf{X}_{2\text{pred}}, \mathbf{X}_{4\text{pred}} \in \mathbb{R}^{10m \times n}$ for $\mathbf{X}_{2\text{train}}, \mathbf{X}_{3\text{train}}$ respectively, and the second objective is to then forecast the future state, matrices $\mathbf{X}_{3\text{pred}}, \mathbf{X}_{5\text{pred}} \in \mathbb{R}^{m \times n}$ for $\mathbf{X}_{2\text{train}}, \mathbf{X}_{3\text{train}}$ respectively. For the first task, a least-square fit is used between the approximation of the denoised data and the truth, and for the forecasting a long-time evaluation is computed leading to the following scores:

$$E_3 = 100(1 - S_{\text{ST}}(\mathbf{X}_{2\text{pred}}, \mathbf{X}_{2\text{test}})), \quad E_4 = 100(1 - S_{\text{LT}}(\mathbf{X}_{3\text{pred}}, \mathbf{X}_{3\text{test}})),$$
$$E_5 = 100(1 - S_{\text{ST}}(\mathbf{X}_{4\text{pred}}, \mathbf{X}_{4\text{test}})), \quad E_6 = 100(1 - S_{\text{LT}}(\mathbf{X}_{5\text{pred}}, \mathbf{X}_{5\text{test}})).$$

**Input:** $\mathbf{X}_{2\text{train}}, \mathbf{X}_{3\text{train}} \in \mathbb{R}^{10m \times n}$;  **Output:** $\mathbf{X}_{2\text{pred}}, \mathbf{X}_{4\text{pred}} \in \mathbb{R}^{10m \times n}$, $\mathbf{X}_{3\text{pred}}, \mathbf{X}_{5\text{pred}} \in \mathbb{R}^{m \times n}$; **Scores:** $E_3, E_4, E_5, E_6$.

### 2.1.3 Test 3: Limited Data (4 scores)

Data limitations are common in real world physical systems and often affect the success of data-driven methods. Thus, testing for model performance on low-data is critically important and provides important insight to potential users.

Figure 2-e demonstrates the nature of the test. In this case only a limited number of snapshots $M$ on numerically accurate data are given $\mathbf{X}_{4\text{train}} \in \mathbb{R}^{M \times n}$. From this limited data, a forecast must be made which is evaluated with both error metrics (2) & (3) on the approximated future $\mathbf{X}_{6\text{pred}} \in \mathbb{R}^{m \times n}$. The experiment is repeated with noise on the measurements using the training matrix $\mathbf{X}_{5\text{train}} \in \mathbb{R}^{M \times n}$ for which a forecasting prediction matrix is produced $\mathbf{X}_{7\text{pred}} \in \mathbb{R}^{m \times n}$. The performance is evaluated on the following scores representing short and long-time metrics for both noise-free and noisy data respectively.

$$E_7 = 100(1 - S_{\text{ST}}(\mathbf{X}_{6\text{pred}}, \mathbf{X}_{6\text{test}})), \quad E_8 = 100(1 - S_{\text{LT}}(\mathbf{X}_{6\text{pred}}, \mathbf{X}_{6\text{test}})),$$
$$E_9 = 100(1 - S_{\text{ST}}(\mathbf{X}_{7\text{pred}}, \mathbf{X}_{7\text{test}})), \quad E_{10} = 100(1 - S_{\text{LT}}(\mathbf{X}_{7\text{pred}}, \mathbf{X}_{7\text{test}})).$$

Two error scores (analogous to $E_1$ and $E_2$) are produced for the noise-free and noisy limited data. These scores are $E_7$ (short) and $E_8$ (long) for the noise free case and $E_9$ (short) and $E_{10}$ (long) for the noisy case.

**Input:** $\mathbf{X}_{4\text{train}}, \mathbf{X}_{5\text{train}} \in \mathbb{R}^{M \times n}$;  **Output:** $\mathbf{X}_{6\text{pred}}, \mathbf{X}_{7\text{pred}} \in \mathbb{R}^{m \times n}$;  **Scores:** $E_7, E_8, E_9, E_{10}$.

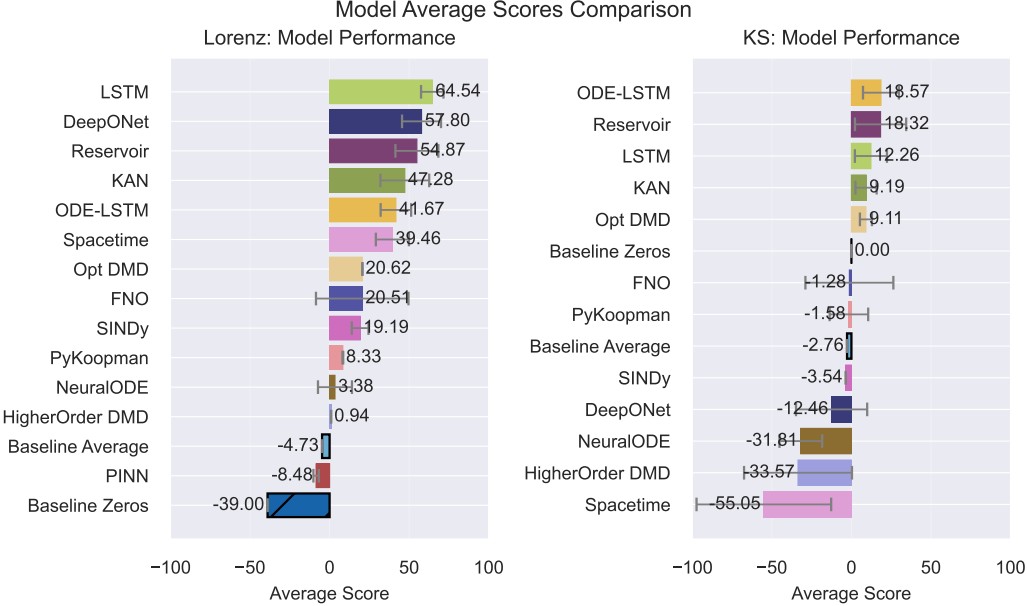

Figure 3: Ranked average scores of each model on the KS and Lorenz Dataset.

#### 2.1.4 Test 4: Parametric Generalization (2 scores)

Finally, the ability of a model to generalize to different parameter values is evaluated. For this case, the model's ability to interpolate and extrapolate to new parameter regimes is considered with noise-free data. The interpolation and extrapolation are each their own score, resulting in two scores that evaluate parametric dependence.

Figure 2-f shows the basic architecture of the test. For the noise-free case, three training data sets are provided with three different (unknown) parameter values $\mathbf{X}_{6\text{train}}, \mathbf{X}_{7\text{train}}, \mathbf{X}_{8\text{train}} \in \mathbb{R}^{10m \times n}$. Construction of the dynamics in parametric regimes that are interpolatory $\mathbf{X}_{8\text{pred}} \in \mathbb{R}^{m \times n}$ and extrapolatory $\mathbf{X}_{9\text{pred}} \in \mathbb{R}^{m \times n}$ are required. For both of the test regimes, a burn in matrix $\mathbf{X}_{9\text{train}}$ and $\mathbf{X}_{10\text{train}}$ respectively of size $M \times n$ is given and the performance is evaluated using the short term metric (2).

$$E_{11} = 100(1 - S_{\text{ST}}(\mathbf{X}_{8\text{pred}}, \mathbf{X}_{8\text{test}})), \quad E_{12} = 100(1 - S_{\text{ST}}(\mathbf{X}_{9\text{pred}}, \mathbf{X}_{9\text{test}})).$$

**Input:** $\mathbf{X}_{6\text{train}}, \mathbf{X}_{7\text{train}}, \mathbf{X}_{8\text{train}} \in \mathbb{R}^{10m \times n}, \mathbf{X}_{9\text{train}}, \mathbf{X}_{10\text{train}} \in \mathbb{R}^{M \times n}$;

**Output:** $\mathbf{X}_{8\text{pred}}, \mathbf{X}_{9\text{pred}} \in \mathbb{R}^{m \times n}$;   **Scores:** $E_{11}, E_{12}$.

### 2.2 Dynamical System: Lorenz

One of the most influential dynamical systems in history, the Lorenz dynamical system is given by

$$\frac{dx}{dt} = \sigma(y - x), \quad \frac{dy}{dt} = rx - xz - y, \quad \frac{dz}{dt} = xy - bz.$$

where the parameters $b = 8/3$ and $\sigma = 10$ are typically fixed at these values while $r$ is explored as a bifurcation parameter. For specific values of $r$, including our choice $r = 28$, the system exhibits chaotic behavior as shown in Fig. 2(a).

Table 1: Model performances for each metric on each dataset (mean ± std).

| Model | Avg Score | E1 | E2 | E3 | E4 | E5 | E6 | E7 | E8 | E9 | E10 | E11 | E12 |
|---|---|---|---|---|---|---|---|---|---|---|---|---|---|
| LSTM [13] | 64.54 (± 0.00) | 99.34 (± 0.18) | 52.37 (± 12.34) | 97.42 (± 0.10) | 50.75 (± 28.41) | **96.44 (± 0.10)** | 72.13 (± 3.54) | 66.61 (± 6.43) | 32.75 (± 16.91) | 36.39 (± 2.05) | 29.57 (± 14.47) | 80.58 (± 0.00) | 60.11 (± 0.00) |
| DeepONet [21] | 57.80 (± 0.00) | 99.11 (± 0.27) | 45.28 (± 40.54) | 96.23 (± 0.00) | 62.32 (± 5.49) | 91.84 (± 0.71) | 14.35 (± 43.11) | 21.84 (± 5.71) | 40.75 (± 18.96) | 34.53 (± 2.52) | 25.15 (± 10.17) | 85.52 (± 5.57) | 76.68 (± 13.97) |
| Reservoir [14, 22, 27] | 54.87 (± 0.00) | **99.91 (± 0.06)** | 56.72 (± 10.00) | 97.41 (± 0.01) | -33.49 (± 70.39) | 93.00 (± 0.01) | 95.07 (± 0.00) | 18.99 (± 47.02) | 65.21 (± 0.84) | 45.41 (± 0.84) | 48.51 (± 24.31) | 41.69 (± 38.67) | 60.85 (± 0.17) |
| KAN [20] | 47.28 (± 0.00) | 82.89 (± 26.43) | 20.53 (± 58.61) | 96.19 (± 0.06) | 55.20 (± 3.28) | 93.00 (± 0.01) | 66.69 (± 5.20) | 50.52 (± 8.64) | 61.92 (± 9.21) | 33.68 (± 4.41) | -31.47 (± 38.67) | **99.80 (± 0.12)** | **99.97 (± 0.00)** |
| ODE-LSTM [6] | 41.67 (± 0.00) | 97.77 (± 0.95) | 17.07 (± 48.03) | **97.90 (± 0.14)** | **69.92 (± 2.74)** | **96.48 (± 0.08)** | 55.82 (± 8.48) | 15.20 (± 28.53) | 36.83 (± 28.53) | 14.56 (± 0.00) | 39.90 (± 0.00) | 40.95 (± 0.00) | 54.09 (± 0.00) |
| Spacetime [36] | 39.46 (± 0.00) | 19.26 (± 15.01) | 83.52 (± 15.01) | 46.79 (± 0.01) | 12.16 (± 66.22) | 56.00 (± 8.78) | 28.28 (± 34.66) | 23.68 (± 0.00) | 33.12 (± 0.00) | 0.00 (± 0.00) | 77.32 (± 0.00) | 59.23 (± 0.00) | 54.09 (± 0.00) |
| OptDMD [1] | 20.62 (± 0.00) | 52.08 (± 0.00) | -67.33 (± 0.00) | 55.51 (± 0.00) | 1.33 (± 0.00) | -64.13 (± 0.00) | 59.11 (± 0.00) | 8.67 (± 0.00) | 42.15 (± 0.00) | -15.73 (± 0.00) | 59.68 (± 0.00) | 59.23 (± 0.00) | 54.09 (± 0.00) |
| FNO [19] | 20.51 (± 0.00) | 50.88 (± 12.55) | -33.65 (± 75.34) | 54.69 (± 0.00) | -9.47 (± 63.22) | 56.40 (± 0.00) | 20.82 (± 16.51) | 51.36 (± 6.19) | 51.36 (± 6.19) | **32.53 (± 19.61)** | 29.29 (± 8.73) | 57.95 (± 11.03) | 57.95 (± 11.03) |
| SINDy [3, 11] | 19.19 (± 0.00) | 81.83 (± 0.00) | 36.00 (± 0.00) | 36.85 (± 0.00) | -29.22 (± 0.00) | -18.27 (± 0.00) | 55.82 (± 8.48) | 15.20 (± 28.53) | 36.83 (± 28.53) | 14.56 (± 23.22) | 82.38 (± 3.22) | 82.38 (± 3.22) | 15.07 (± 0.00) |
| PyKoopman [2, 26] | 8.33 (± 0.00) | 34.50 (± 0.00) | **89.87 (± 0.00)** | 54.97 (± 0.01) | 52.40 (± 0.00) | -90.59 (± 0.72) | -22.31 (± 0.00) | -93.73 (± 0.00) | 43.93 (± 0.00) | -78.67 (± 0.00) | 25.63 (± 0.00) | 27.49 (± 0.00) | 27.49 (± 0.00) |
| NeuralODE [5] | 3.38 (± 0.00) | 43.40 (± 7.99) | -40.37 (± 21.82) | 53.88 (± 0.76) | -14.75 (± 14.88) | -36.16 (± 12.98) | 45.61 (± 11.22) | -83.55 (± 11.22) | 32.93 (± 18.26) | -85.20 (± 18.26) | 31.35 (± 13.08) | 38.03 (± 14.49) | 38.03 (± 14.49) |
| HigherOrder DMD [18] | 0.94 (± 0.00) | 51.77 (± 0.00) | -84.40 (± 0.00) | 54.88 (± 0.00) | -91.87 (± 0.00) | -90.80 (± 0.00) | **66.85 (± 0.00)** | -81.60 (± 0.00) | 49.74 (± 0.00) | -11.33 (± 0.00) | 59.04 (± 0.00) | 31.22 (± 0.00) | 31.22 (± 0.00) |
| Baseline Average | -4.73 (± 0.00) | 51.71 (± 0.00) | -91.20 (± 0.00) | 54.88 (± 0.00) | -88.67 (± 1.50) | -91.33 (± 0.00) | 66.97 (± 0.00) | -91.07 (± 0.00) | **51.93 (± 1.21)** | -90.27 (± 0.00) | 57.08 (± 0.00) | 60.88 (± 0.00) | 60.88 (± 0.00) |
| PINN [28] | -8.48 (± 0.00) | 62.77 (± 0.25) | -91.47 (± 0.00) | 55.83 (± 0.09) | -88.67 (± 0.00) | -89.47 (± 0.00) | 23.64 (± 0.64) | -94.53 (± 7.70) | 45.36 (± 8.33) | -96.40 (± 0.56) | 53.02 (± 0.01) | 61.64 (± 0.02) | 61.64 (± 0.02) |
| Baseline Zeros | -39.00 (± 0.00) | 0.00 (± 0.00) | -93.33 (± 0.00) | 0.00 (± 0.00) | -93.47 (± 0.00) | -93.73 (± 0.00) | 0.00 (± 0.00) | -93.73 (± 0.00) | 0.00 (± 0.00) | -93.73 (± 0.00) | 0.00 (± 0.00) | 0.00 (± 0.00) | 0.00 (± 0.00) |

(a) Model Scores on Lorenz Dataset

| Model | Avg Score | E1 | E2 | E3 | E4 | E5 | E6 | E7 | E8 | E9 | E10 | E11 | E12 |
|---|---|---|---|---|---|---|---|---|---|---|---|---|---|
| ODE-LSTM [6] | 18.57 (± 0.00) | 80.09 (± 0.27) | 15.68 (± 35.88) | 88.65 (± 0.06) | 26.41 (± 6.46) | 52.18 (± 0.42) | -47.57 (± 48.94) | 1.71 (± 7.42) | **57.95 (± 5.34)** | **6.37 (± 3.63)** | 8.19 (± 2.41) | -54.07 (± 0.00) | -12.76 (± 24.73) |
| Reservoir [14, 22, 27] | 18.32 (± 0.00) | **99.97 (± 0.00)** | **86.45 (± 1.75)** | 88.61 (± 0.04) | 18.52 (± 14.25) | **80.73 (± 0.06)** | -21.95 (± 14.05) | -12.38 (± 6.39) | 7.43 (± 18.11) | -100.00 (± 30.40) | -100.00 (± 100.00) | **32.39 (± 4.16)** | **40.08 (± 3.93)** |
| LSTM [13] | 12.26 (± 0.00) | 95.22 (± 0.61) | 4.78 (± 15.15) | **90.11 (± 0.01)** | -23.63 (± 38.59) | 79.83 (± 0.07) | -98.11 (± 35.17) | **7.28 (± 1.75)** | 49.86 (± 6.25) | 4.45 (± 7.68) | 31.70 (± 15.72) | -54.07 (± 0.00) | -40.31 (± 0.00) |
| KAN [20] | 9.19 (± 0.00) | -4.43 (± 1.11) | 16.74 (± 1.16) | 50.36 (± 0.86) | 2.17 (± 2.45) | 36.93 (± 1.20) | 10.17 (± 7.54) | -22.46 (± 13.79) | 32.97 (± 19.49) | -43.06 (± 15.77) | 32.83 (± 13.72) | 0.83 (± 0.17) | -2.75 (± 2.67) |
| Opt DMD [1] | 9.11 (± 0.00) | 53.36 (± 0.57) | 15.58 (± 0.01) | 6.90 (± 0.02) | 6.81 (± 0.04) | 8.82 (± 0.17) | 13.49 (± 5.08) | -11.10 (± 0.00) | 24.57 (± 0.00) | -71.97 (± 28.68) | **55.70 (± 11.30)** | 6.00 (± 0.00) | 1.12 (± 0.01) |
| Baseline Zeros | 0.00 (± 0.00) | 0.00 (± 0.00) | 0.00 (± 0.00) | 0.00 (± 0.00) | 0.00 (± 0.00) | 0.00 (± 0.00) | 0.00 (± 0.00) | 0.00 (± 0.00) | 0.00 (± 0.00) | 0.00 (± 0.00) | 0.00 (± 0.00) | 0.00 (± 0.00) | 0.00 (± 0.00) |
| FNO [19] | -1.28 (± 0.00) | 99.00 (± 0.23) | -100.00 (± 100.00) | 87.44 (± 0.00) | **74.46 (± 9.09)** | 75.46 (± 0.00) | **54.34 (± 8.09)** | -8.66 (± 9.03) | -100.00 (± 100.00) | -6.24 (± 5.25) | -95.43 (± 100.00) | -54.62 (± 0.00) | -41.15 (± 0.00) |
| PyKoopman [2, 26] | -1.58 (± 0.00) | 14.60 (± 0.57) | 38.58 (± 7.45) | 0.00 (± 0.00) | 24.35 (± 24.68) | 0.01 (± 0.00) | 25.37 (± 0.00) | -17.37 (± 3.36) | -100.00 (± 100.00) | -6.90 (± 0.00) | 0.02 (± 0.00) | 2.41 (± 9.30) | 0.02 (± 0.06) |
| Baseline Average | -2.76 (± 0.00) | -3.39 (± 0.00) | 4.34 (± 0.00) | 0.01 (± 0.00) | 0.13 (± 0.00) | 0.40 (± 0.00) | 0.22 (± 0.00) | -9.23 (± 0.00) | 7.66 (± 0.00) | -7.12 (± 0.00) | 15.67 (± 0.00) | -27.97 (± 0.00) | -13.88 (± 0.00) |
| SINDy [3, 11] | -3.54 (± 0.00) | 84.38 (± 0.00) | -1.82 (± 0.00) | -2.91 (± 0.00) | -92.67 (± 0.00) | -1.23 (± 0.00) | -100.00 (± 0.00) | -0.22 (± 0.00) | 31.72 (± 0.00) | -14.00 (± 0.00) | 33.79 (± 0.00) | 10.01 (± 0.00) | 10.51 (± 0.00) |
| DeepONet [21] | -12.46 (± 0.00) | 36.52 (± 3.85) | 9.94 (± 21.38) | -1.45 (± 17.51) | 6.77 (± 8.68) | 6.29 (± 1.42) | -100.00 (± 100.00) | -9.48 (± 4.27) | -100.00 (± 100.00) | -1.93 (± 0.00) | -0.30 (± 0.06) | -4.60 (± 7.22) | 8.77 (± 4.07) |
| NeuralODE [5] | -31.81 (± 0.00) | -36.06 (± 15.82) | -9.43 (± 29.99) | -100.00 (± 11.75) | 10.09 (± 6.83) | -100.00 (± 9.21) | 27.03 (± 9.92) | -56.98 (± 6.69) | -44.19 (± 51.95) | -98.34 (± 8.54) | 13.99 (± 9.70) | 2.05 (± 0.23) | 10.13 (± 0.22) |
| HigherOrder DMD [18] | -33.57 (± 0.00) | -100.00 (± 100.00) | -100.00 (± 100.00) | 0.00 (± 0.00) | 0.00 (± 0.00) | 0.00 (± 0.00) | 0.00 (± 0.00) | -3.22 (± 6.91) | -100.00 (± 100.00) | -0.03 (± 0.07) | -100.00 (± 100.00) | 0.00 (± 0.00) | 0.46 (± 0.00) |
| Spacetime [36] | -55.05 (± 0.00) | 43.49 (± 100.00) | -100.00 (± 100.00) | -42.95 (± 0.00) | -100.00 (± 0.00) | -43.20 (± 0.00) | -100.00 (± 0.00) | -35.56 (± 7.78) | -100.00 (± 100.00) | -57.17 (± 0.00) | -100.00 (± 100.00) | -31.28 (± 0.00) | 6.04 (± 0.00) |

(b) Model Scores on Kuramoto–Sivashinsky Dataset

The training and testing are identical as for the spatio-temporal KS system described above aside from the long range (climate) forecast score. Data matrices for testing and training are of the same form as in Section 2.1 with $n = 3$ being the dimension of the dynamical system. Since in this case there is no spatial coordinate it is no longer possible to use the power spectral density of the differential equation to evaluate the long-time performance. Instead, for this system, we evaluate the long-time forecasting based on the distribution of values in the state-space over the last $k$ time steps (e.g. $k = 500$). For this we compare the histograms of the distribution of predicted and true solution trajectories in the following way. The histogram for a time series is computed using the histogram command with a set number of bins (e.g., $bins = 41$ for our current Lorenz evaluation). The difference of the histogram between the truth ($x$, $y$ and $z$) and prediction ($\tilde{x}$, $\tilde{y}$ and $\tilde{z}$) for each variable is measured in an $\ell_1$-sense:

$$s_{\text{LT}}(x, \tilde{x}) = \frac{\|\text{Hist}_x - Hist_{\tilde{x}}\|_1}{\|\text{Hist}\|_1}.$$

From this the long-time error score for the Lorenz system is composed of the distributional error in each coordinate:

$$S_{\text{LT}}^{(\text{Lorenz})}(\mathbf{X}, \tilde{\mathbf{X}}) = (s_{\text{LT}}(x, \tilde{x}) + s_{\text{LT}}(y, \tilde{y}) + s_{\text{LT}}(z, \tilde{z}))/3 \qquad \text{(climate forecast)}.$$

As with the spatio-temporal system and the power spectral density, this gives a simple measure of the accuracy of the prediction from a statistical viewpoint since long-time prediction is well beyond the Lyapunov time which would not allow for a least-square match between trajectories of the truth and prediction.

### 2.3   Composite Score

We compute a composite score $\bar{E}$ per dataset from metrics $E_1$ through $E_{12}$ by averaging the resulting scores for each method. This score is evaluated per method, not per model. Thus, each method can fit a model for each task and produce the best possible score. All scores are clipped such that $E_i \in [-100, 100]$, thus $\bar{E} \in [-100, 100]$. Methods that cannot produce a result for a given task receive the minimum score $-100$.

## 3   Methods, Baselines and Results

We characterized twelve highly-cited modeling methods on our **ctf4science** datasets. Table 1 shows all scored methods and their resulting performance scores. For details on the scored methods, as well as the hyperparameter tuning and evaluation procedures, please refer to the appendix. In addition, we also provide the scores of six zero-shot time-series forecasting foundation models in Table 5 of the appendix. The **ctf4science** includes two naive baseline methods: predicting zero and predicting the average. In our evaluations, we use average prediction as the baseline for the Lorenz dataset and zero prediction as the reference baseline for KS dataset.

In Fig. 3, we show all evaluated methods per dataset including the naive baselines—constant and average—ranked by their $\bar{E}$. The difference in dimensionality, dynamics, and long-term trajectory stability between Lorenz and KS results in radically different performance distributions. Further, while some models score high on specific tasks, no model scores high-across all tasks (see Table 1). Overall, the results demonstrate that each dataset and task is challenging enough to produce a distribution of scores that characterizes the methods.

A complete overview of all model's performance metrics on the Lorenz dataset can be found in table 1a. The overall score performance for each method in in Fig. 3 while the top three performers in each error category are shown is shown in Fig. 4(a). A complete overview of all model's performance metrics on the KS dataset can be found in table 1b. The overall score performance for each method in in Fig. 3 while the top three performers in each error category are shown is shown in Fig. 4(b).

### 3.1   Observations

Applying the "ImageNet recipe" (fixed public data, objective metrics, leaderboarded methods) to dynamic systems poses new challenges. Scientific models are not trivial to compare, as they range from assumption-rich, high-fidelity approaches to generic, assumption-free, data-hungry models. While the low-dimensional chaotic Lorenz ODE is canonical, easy to synthesize, and analytically

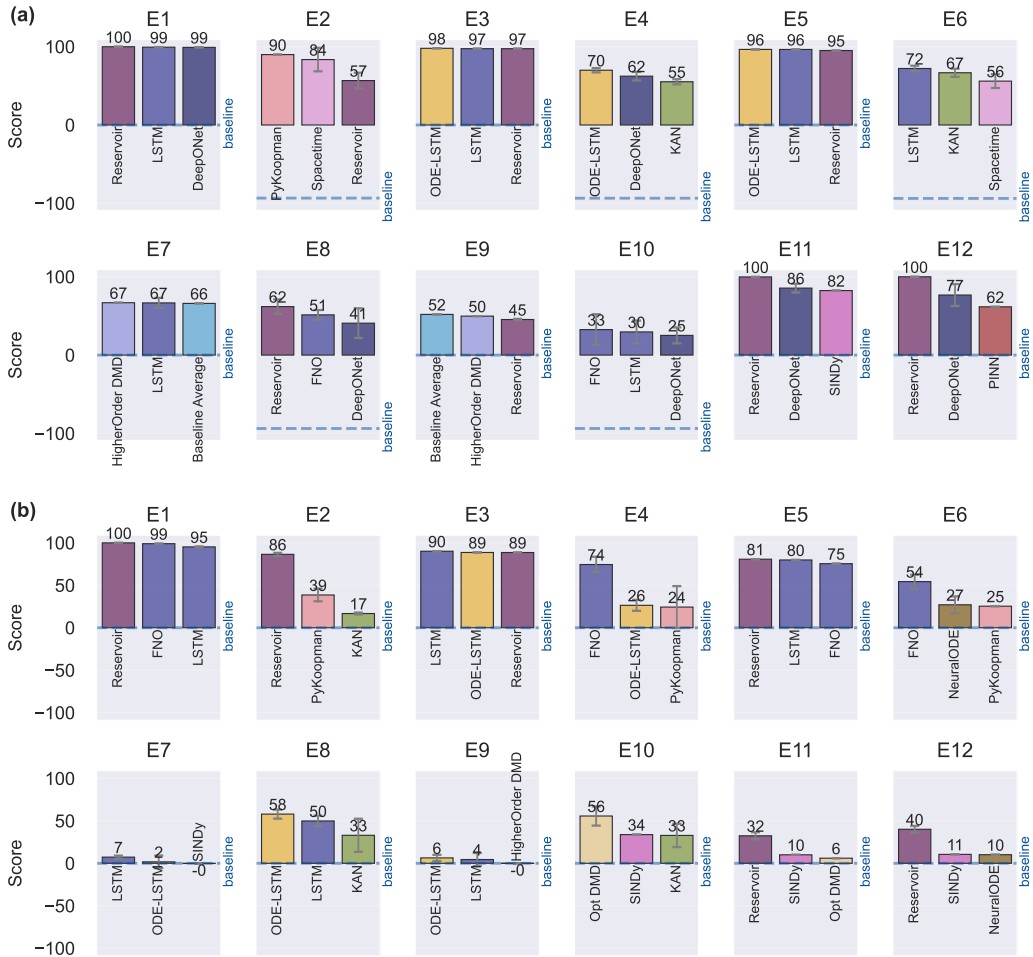

Figure 4: Top three performing models per metric on the (a) Lorenz and (b) KS dataset. The blue baseline line here corresponds to the constant zero prediction. This baseline is not producing a score of zero in long-time predictions for the Lorenz dataset due to the different long-time evaluation methods used for KS and Lorenz. KS uses spectral $L_2$-error whereas Lorenz uses histogram $L_2$-error.

transparent, it is chaotic. Chaos guarantees that any forecaster—even the ground-truth solver—accumulates exponential error beyond 3 Lyapunov times, so "predict-the-mean" becomes the rational long-horizon baseline.

Methods therefore succeed or fail depending on whether their implicit assumptions match the task: SINDy excels when its candidate library contains the true terms; operator learners and PINNs might under-perform because they were designed for smooth function-to-function or interpolation problems, not autoregressive time marching; generic RNN-style models struggle at the low data limit, while reservoir models are very well adapted for chaotic time series. Simultaneously, we also see some methods unexpectedly outperformed others in contexts they were not designed for (e.g., DeepONet applied to an autoregressive task on temporal, rather than spatio-temporal data). In essence, **ctf4science** works as intended. Every task-dataset combination acts as a search light illuminating the performance space within which modeling methods exist and provide insight into which method can tackle which under which conditions.

We begin by presenting a ranking of all methods evaluated from their composite score (See Fig. 3 and Table 1). We present the top 3 models and the constant prediction baseline for each metric from $E1$ through $E12$. The results highlight how the diversity of methods developed have definitive strengths and weaknesses on the various tasks. Thus depending on the task, the appropriate method should be deployed. The CTF provides the critical evaluation metrics necessary for making such decisions.

# 4    Limitations & Future Work

We are launching **ctf4science** in a limited scope with three datasets: a dynamical system (Lorenz) and two spatio-temporal system (KS and SST). The evaluation metrics test short- and long-time forecasting and reconstruction under the challenges of noise, limited data and parametric dependency. There are many more datasets and tasks that could and should be considered for science and engineering, most notably tasks in control. This CTF is an important first step to establish fair comparisons among modeling methods on truly withheld test sets. In future versions, more challenging datasets, real world datasets, and more tasks, including control tasks will be integrated.

A key limiting factor in achieving high-scores on the current CTF datasets is the small dataset size, which hamstrings large machine learning models from performing at their best. This was by design, since in many engineering systems, limited data availability is a practical reality. We will expand our collection of datasets and scoring metrics to larger datasets in the future.

Furthermore, the current selection of models is only a starting point. We fully expect that extensions to standard methods could outperform our results (e.g. PINNs[34]). We want to improve on the current results together with the broader research community. **ctf4science** will help us find successful variations and new applications to existing methods.

While wall-clock time is a useful metric for assessing the potential speed advantage of ML methods over traditional approaches[23], our focus here is on evaluating model suitability for certain tasks. Wall-clock time depends on factors such as hardware configuration, implementation, parallelization, and library efficiency. Nevertheless, we provide our time measurement of each model's training and evaluation pipeline in the appendix (Table 4) as a rough indication of computational burden.

# 5    Conclusion

We developed a CTF that scores modeling approaches on a diversity of tasks that are prototypical in science and engineering. The canonical Lorenz and KS systems form an accepted testbench for demonstrating the effectiveness of modeling methods in scientific machine learning literature and act as the starting point of our benchmark. Our work builds a fair and multi-dimensional comparison between methods that is based on a true hidden test set—limiting the risk of "hacked" scores.

CTFs have transformed the research fields that embraced them, such as computer vision, speech and language processing. CTFs have also been critical in identifying protein structure from sequence [16], leading to the Nobel Prize in Chemistry. Scientific machine learning is now mature enough as a field that a CTF is warranted and needed in order to fairly and accurately evaluate emerging algorithms, especially on the diversity of tasks critical to science and engineering. This work marks the beginning of a sustained effort to provide a neutral and fair comparison between methods and tasks, and thereby boost transparency and competition in machine learning for science.

The central tension our experiment exposes is that scientific ML methods live on a spectrum from assumption-rich, high fidelity to generic, assumption-free, data-hungry models. We see the present CTF as the microscope slide on which this spectrum first becomes visible. Our roadmap adds diverse systems (non-chaotic ODEs, PDEs, stochastic SDEs, experimental datasets), multiple task types (forecasting, system identification, imputation, control), and configuration files that declare what priors each submission may exploit. By exposing where and why celebrated learning algorithms misalign with specific scientific goals, the current CTF is not a verdict on their value but an invitation to researchers in the community to refine architectures and to co-create a truly comprehensive benchmark suite for scientific machine learning; enabling the discovery of scientific breakthroughs and foundational world models.

## Acknowledgments and Disclosure of Funding

The authors acknowledge support from the National Science Foundation AI Institute in Dynamic Systems (grant number 2112085). GM acknowledges support from the EPSRC programme grant in 'The Mathematics of Deep Learning' (project EP/V026259/1). The hyperparameter tuning and final evaluation of all models were carried out on the Dutch national supercomputer Snellius, provided by SURF. AY is supported by the NSF GRFP (No. DGE-2140004).

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
