# Supplement: Common Task Framework For a Critical Evaluation of Scientific Machine Learning Algorithms

Philippe M. Wyder[1],        Judah Goldfeder[2],        Alexey Yermakov[1,3],        Yue Zhao[4],

Stefano Riva[6],        Jan Williams[5],        David Zoro[3],        Amy Sara Rude[1],

Matteo Tomasetto[7],        Joe Germany[8],        Joseph Bakarji[9],        Georg Maierhofer[10],

Miles Cranmer[10],                        J. Nathan Kutz[1,3] *

[1]Department of Applied Mathematics, University of Washington, Seattle, WA 98195
[2]Department of Computer Science, Columbia University, New York, NY 10027
[3]Department of Electrical and Computer Engineering, University of Washington, Seattle, WA 98195
[4]High Performance Machine Learning, SURF, Amsterdam, the Netherlands
[5]Department of Mechanical Engineering, University of Washington, Seattle, WA 98195
[6]Department of Energy, Nuclear Engineering Division, Politecnico di Milano, Milan, Italy
[7]Department of Mechanical Engineering, Politecnico di Milano, Milan, Italy
[8]Department of Mathematics, American University in Beirut, Beirut, Lebanon
[9]Department of Mechanical Engineering, American University in Beirut, Beirut, Lebanon
[10]Department of Applied Mathematics and Theoretical Physics, University of Cambridge, Cambridge, UK

*Corresponding author: `kutz@uw.edu`

39th Conference on Neural Information Processing Systems (NeurIPS 2025) Track on Datasets and Benchmarks.

## Overview

This document contains the supplementary materials for the *Common Task Framework For a Critical Evaluation of Scientific Machine Learning Algorithms* paper. For each model that was evaluated on the CTF4Science, we share additional implementation and hyperparameter tuning details. This document assumes familiarity with the main text and thus does not redefine terms and details covered in the main text, such as the scoring metrics $E1 - E12$.

## Contents

# 1 Dataset Files and Evaluation Metrics

Table 1: Files and corresponding evaluation metrics ($E_1$–$E_{12}$) for benchmark datasets.

| Score | Test | Task | Train / Burn-in File(s) | Ground Truth File |
|---|---|---|---|---|
| $E_1$ | Forecasting | Short-time | $\mathbf{X}_{1\text{train}}$ | $\mathbf{X}_{1\text{test}}$ |
| $E_2$ | Forecasting | Long-time | $\mathbf{X}_{1\text{train}}$ | $\mathbf{X}_{1\text{test}}$ |
| $E_3$ | Noisy (medium) | Reconstruction (denoising) | $\mathbf{X}_{2\text{train}}$ | $\mathbf{X}_{2\text{test}}$ |
| $E_4$ | Noisy (medium) | Forecast (long-time) | $\mathbf{X}_{2\text{train}}$ | $\mathbf{X}_{3\text{test}}$ |
| $E_5$ | Noisy (high) | Reconstruction (denoising) | $\mathbf{X}_{3\text{train}}$ | $\mathbf{X}_{4\text{test}}$ |
| $E_6$ | Noisy (high) | Forecast (long-time) | $\mathbf{X}_{3\text{train}}$ | $\mathbf{X}_{5\text{test}}$ |
| $E_7$ | Limited Data (clean) | Forecast (short-time) | $\mathbf{X}_{4\text{train}}$ | $\mathbf{X}_{6\text{test}}$ |
| $E_8$ | Limited Data (clean) | Forecast (long-time) | $\mathbf{X}_{4\text{train}}$ | $\mathbf{X}_{6\text{test}}$ |
| $E_9$ | Limited Data (noisy) | Forecast (short-time) | $\mathbf{X}_{5\text{train}}$ | $\mathbf{X}_{7\text{test}}$ |
| $E_{10}$ | Limited Data (noisy) | Forecast (long-time) | $\mathbf{X}_{5\text{train}}$ | $\mathbf{X}_{7\text{test}}$ |
| $E_{11}$ | Parametric Generalization | Interpolation forecast | $\mathbf{X}_{6,7,8\text{train}}$ / $\mathbf{X}_{9\text{train}}$ | $\mathbf{X}_{8\text{test}}$ |
| $E_{12}$ | Parametric Generalization | Extrapolation forecast | $\mathbf{X}_{6,7,8\text{train}}$ / $\mathbf{X}_{10\text{train}}$ | $\mathbf{X}_{9\text{test}}$ |

Table 2: Matrix shapes and indices for the Lorenz dataset (left) and Kuramoto-Sivashinsky dataset (right). Start and end index refer to relative time-steps in the simulation used to generate the dataset matrices. Each successive index represents one $\Delta t$ time-step.

| Lorenz | | | | | Kuramoto-Sivashinsky | | | |
|---|---|---|---|---|---|---|---|---|
| **Matrix** | **Shape** | **Start Index** | **End Index** | | **Matrix** | **Shape** | **Start Index** | **End Index** |
| $\mathbf{X}_{1\text{train}}$ | $[10000, 3]$ | 0 | 10000 | | $\mathbf{X}_{1\text{train}}$ | $[10000, 1024]$ | 0 | 10000 |
| $\mathbf{X}_{2\text{train}}$ | $[10000, 3]$ | 0 | 10000 | | $\mathbf{X}_{2\text{train}}$ | $[10000, 1024]$ | 0 | 10000 |
| $\mathbf{X}_{3\text{train}}$ | $[10000, 3]$ | 0 | 10000 | | $\mathbf{X}_{3\text{train}}$ | $[10000, 1024]$ | 0 | 10000 |
| $\mathbf{X}_{4\text{train}}$ | $[100, 3]$ | 0 | 100 | | $\mathbf{X}_{4\text{train}}$ | $[100, 1024]$ | 0 | 100 |
| $\mathbf{X}_{5\text{train}}$ | $[100, 3]$ | 0 | 100 | | $\mathbf{X}_{5\text{train}}$ | $[100, 1024]$ | 0 | 100 |
| $\mathbf{X}_{6\text{train}}$ | $[10000, 3]$ | 0 | 10000 | | $\mathbf{X}_{6\text{train}}$ | $[10000, 1024]$ | 0 | 10000 |
| $\mathbf{X}_{7\text{train}}$ | $[10000, 3]$ | 0 | 10000 | | $\mathbf{X}_{7\text{train}}$ | $[10000, 1024]$ | 0 | 10000 |
| $\mathbf{X}_{8\text{train}}$ | $[10000, 3]$ | 0 | 10000 | | $\mathbf{X}_{8\text{train}}$ | $[10000, 1024]$ | 0 | 10000 |
| $\mathbf{X}_{9\text{train}}$ | $[100, 3]$ | 9900 | 10000 | | $\mathbf{X}_{9\text{train}}$ | $[100, 1024]$ | 9900 | 10000 |
| $\mathbf{X}_{10\text{train}}$ | $[100, 3]$ | 9900 | 10000 | | $\mathbf{X}_{10\text{train}}$ | $[100, 1024]$ | 9900 | 10000 |
| $\mathbf{X}_{1\text{test}}$ | $[1000, 3]$ | 10000 | 11000 | | $\mathbf{X}_{1\text{test}}$ | $[1000, 1024]$ | 10000 | 11000 |
| $\mathbf{X}_{2\text{test}}$ | $[10000, 3]$ | 0 | 10000 | | $\mathbf{X}_{2\text{test}}$ | $[10000, 1024]$ | 0 | 10000 |
| $\mathbf{X}_{3\text{test}}$ | $[1000, 3]$ | 10000 | 11000 | | $\mathbf{X}_{3\text{test}}$ | $[1000, 1024]$ | 10000 | 11000 |
| $\mathbf{X}_{4\text{test}}$ | $[10000, 3]$ | 0 | 10000 | | $\mathbf{X}_{4\text{test}}$ | $[10000, 1024]$ | 0 | 10000 |
| $\mathbf{X}_{5\text{test}}$ | $[1000, 3]$ | 10000 | 11000 | | $\mathbf{X}_{5\text{test}}$ | $[1000, 1024]$ | 10000 | 11000 |
| $\mathbf{X}_{6\text{test}}$ | $[1000, 3]$ | 100 | 1100 | | $\mathbf{X}_{6\text{test}}$ | $[1000, 1024]$ | 100 | 1100 |
| $\mathbf{X}_{7\text{test}}$ | $[1000, 3]$ | 100 | 1100 | | $\mathbf{X}_{7\text{test}}$ | $[1000, 1024]$ | 100 | 1100 |
| $\mathbf{X}_{8\text{test}}$ | $[1000, 3]$ | 10000 | 11000 | | $\mathbf{X}_{8\text{test}}$ | $[1000, 1024]$ | 10000 | 11000 |
| $\mathbf{X}_{9\text{test}}$ | $[1000, 3]$ | 10000 | 11000 | | $\mathbf{X}_{9\text{test}}$ | $[1000, 1024]$ | 10000 | 11000 |

# 2 Evaluations

## 2.1 Hyperparameter Optimization

Hyperparameter optimization is performed in our **ctf4science** Python package[2] using the `tune_module.py` script. We employ Ray Tune [41] for systematic hyperparameter optimization across all models. Hyperparameters are defined in YAML configuration files specifying parameter types, bounds, and sampling distributions. Multiple parameter types are supported, including continuous distributions (uniform, log-uniform), discrete distributions (random integer, log-random integer), and categorical choices.

The optimization follows a trial-based approach where each trial randomly samples a hyperparameter configuration from the defined search space. Each trial trains the model using a train/validation split of the original training dataset. The `tune_module.py` script splits the training data into train and validation sets, using the latter exclusively for evaluation. Thus, the test set remains unseen during hyperparameter tuning.

Optimization terminates when either a predefined number of trials or a time budget is reached. We employ ASHA (Asynchronous Successive Halving Algorithm) scheduling [39] for early stopping of poorly performing trials. Resource allocation is automatically managed, distributing trials across available computational resources.

---

[2]Available at `https://github.com/CTF-for-Science/ctf4science`

Table 3: Average model performances for each metric group on each dataset. E1-E6 demonstrate reconstruction and forecasting performance, E7-E10 demonstrate low-data regime performance, and E11-E12 show parametric generalization performance.

| Model | E1-E6 | E7-E10 | E11-E12 |
|---|---|---|---|
| Baseline Zeros | -46.76 (± 0.00) | -46.87 (± 0.00) | 0.00 (± 0.00) |
| Baseline Average | -18.55 (± 0.00) | -15.86 (± 0.00) | 58.98 (± 0.00) |
| Reservoir [30, 51, 57] | 55.77 (± 21.25) | 31.01 (± 8.59) | **99.89 (± 0.06)** |
| KAN [46] | 69.08 (± 15.60) | 12.57 (± 21.60) | 51.27 (± 2.03) |
| HigherOrder DMD [37] | -17.10 (± 0.00) | 5.91 (± 0.00) | 45.13 (± 0.00) |
| OptDMD [4] | 5.72 (± 0.00) | 23.55 (± 0.00) | 59.46 (± 0.00) |
| PyKoopman [7, 56] | 32.94 (± 0.12) | -37.70 (± 0.54) | 26.56 (± 0.00) |
| LSTM [24] | **78.07 (± 7.44)** | **41.33 (± 9.96)** | 70.34 (± 0.00) |
| ODE-LSTM [15] | 49.46 (± 8.66) | 30.60 (± 15.78) | 40.42 (± 0.00) |
| Spacetime [68] | 42.84 (± 15.00) | 21.27 (± 8.66) | 65.70 (± 0.00) |
| DeepONet [47] | 68.19 (± 15.02) | 30.57 (± 9.34) | 81.10 (± 9.77) |
| SINDy [8, 19] | 1.73 (± 0.00) | 30.60 (± 15.78) | 48.73 (± 0.00) |
| FNO [40] | 25.70 (± 31.14) | 1.18 (± 35.58) | 43.62 (± 9.88) |
| NeuralODE [12] | 10.23 (± 9.89) | -22.55 (± 10.32) | 34.69 (± 13.78) |
| PINN [59] | -15.74 (± 0.41) | -30.48 (± 4.45) | 57.33 (± 0.01) |

(a) Average model performances for each metric group on Lorenz Dataset

| Model | E1-E6 | E7-E10 | E11-E12 |
|---|---|---|---|
| Reservoir [30, 51, 57] | **58.72 (± 5.03)** | -51.24 (± 38.72) | **36.23 (± 4.04)** |
| ODE-LSTM [15] | 35.90 (± 15.34) | 18.55 (± 4.70) | -33.42 (± 12.36) |
| Opt DMD [4] | 17.50 (± 0.89) | -0.70 (± 10.00) | 3.56 (± 0.01) |
| KAN [46] | 18.66 (± 2.39) | 0.07 (± 15.69) | -0.96 (± 1.42) |
| SINDy [8, 19] | -19.04 (± 0.00) | 12.82 (± 0.00) | 10.26 (± 0.00) |
| LSTM [24] | 24.70 (± 14.93) | **23.32 (± 7.85)** | -47.19 (± 0.00) |
| Baseline Zeros | 0.00 (± 0.00) | 0.00 (± 0.00) | 0.00 (± 0.00) |
| PyKoopman [7, 56] | 17.15 (± 5.45) | -31.06 (± 25.84) | 1.21 (± 4.68) |
| Baseline Average | 0.28 (± 0.00) | 1.75 (± 0.00) | -20.92 (± 0.00) |
| DeepONet [47] | -6.99 (± 25.47) | -27.93 (± 26.08) | 2.08 (± 5.65) |
| FNO [40] | 48.45 (± 19.57) | -52.58 (± 53.57) | -47.88 (± 0.00) |
| NeuralODE [12] | -34.73 (± 13.92) | -46.38 (± 19.22) | 6.09 (± 0.22) |
| HigherOrder DMD [37] | -33.33 (± 33.33) | -50.81 (± 51.75) | 0.23 (± 0.00) |
| Spacetime [68] | -57.11 (± 50.00) | -73.18 (± 51.95) | -12.62 (± 0.00) |

(b) Average model performances for each metric group on KS Dataset

For our results, each combination of model, dataset, and pair_id is allocated 8 hours of tuning time on dedicated nodes equipped with 1 NVIDIA A100 GPU with 40 GiB VRAM and 18 CPU cores from an Intel Xeon Platinum 8360Y processor with 120GiB RAM. Some models complete tuning in less than the alotted time.

## 2.2 Evaluation

Model evaluation is performed using our **ctf4science** Python package[3]'s `benchmark_module.py` script. Once hyperparameter tuning is complete, the best-performing parameters on the validation set are used to retrain the model on the full training dataset. The retraining and subsequent evaluation on the test dataset are repeated five times, using different random seeds where possible. We report the mean and standard deviation of the resulting scores across these five runs as indicators of model stability. For models that do not rely on random seeds, the standard deviation is zero. Reported standard deviation values are clipped to a maximum of 100.

## 2.3 Wall-Clock Time

McGreivy and Hakim [53] compared ML methods with traditional approaches under conditions of either equal accuracy or equal runtime, motivated by the claims of the methods in their study that those methods achieve comparable accuracy with improved computational efficiency. In contrast, we take a step back to first examine whether ML methods can achieve reasonable accuracy at all. Therefore, our focus is on the accuracy metrics designed in the paper. Although our goal is not to provide a fair assessment of the speed gain of the ML methods, we nevertheless report the computational costs of the individual models in their current implementations for context. Wall-clock time is measured by our **ctf4science** package's `performance_module.py` script. The total wall-clock time, in seconds,

---

[3]Available at `https://github.com/CTF-for-Science/ctf4science`

required to train and evaluate each model via our package's `run.py` scripts without the visualization option is provided in Table 4.

Table 4: Model mean wall clock times for each pair_id on each dataset

| Model | pair_id 1 | pair_id 2 | pair_id 3 | pair_id 4 | pair_id 5 | pair_id 6 | pair_id 7 | pair_id 8 | pair_id 9 |
|---|---|---|---|---|---|---|---|---|---|
| Baseline Zeros | 0 | 0 | 0 | 0 | 0 | 0 | 0 | 0 | 0 |
| Baseline Average | 0 | 0 | 0 | 0 | 0 | 0 | 0 | 0 | 0 |
| Reservoir [30, 51, 57] | 2 | 12 | 6 | 17 | 2 | 1 | 1 | 17 | 18 |
| KAN [46] | 186 | 134 | 180 | 25 | 1498 | 88 | 85 | 346 | 377 |
| HighOrder DMD [37] | 0 | 0 | 0 | 0 | 0 | 0 | 0 | 0 | 0 |
| OptDMD [4] | 4 | 5 | 5 | 3 | 3 | 0 | 0 | 0 | 0 |
| PyKoopman [7, 56] | 0 | 0 | 0 | 0 | 1 | 0 | 0 | 0 | 0 |
| LSTM [24] | 1377 | 2723 | 146 | 2154 | 1293 | 51 | 54 | 689 | 485 |
| ODE-LSTM [15] | 15667 | 15876 | 12234 | 15057 | 14517 | 231 | 172 | 14447 | 15073 |
| Spacetime [68] | 331 | 832 | 469 | 1187 | 1035 | 28 | 27 | 847 | 744 |
| DeepONet [47] | 234 | 2 | 290 | 39 | 57 | 39 | 40 | 59 | 87 |
| SINDy [8, 19] | 1080 | 937 | 2745 | 3 | 72 | 189 | 70 | 153 | 248 |
| FNO [40] | 417 | 1098 | 924 | 1477 | 375 | 19 | 21 | 907 | 2184 |
| NeuralODE [12] | 9468 | 2172 | 848 | 2390 | 786 | 51 | 27 | 4460 | 3589 |
| PINN [59] | 77 | 77 | 76 | 76 | 76 | 76 | 76 | 76 | 76 |

(a) Mean Wall Clock Times on Lorenz Dataset in Seconds

| Model | pair_id 1 | pair_id 2 | pair_id 3 | pair_id 4 | pair_id 5 | pair_id 6 | pair_id 7 | pair_id 8 | pair_id 9 |
|---|---|---|---|---|---|---|---|---|---|
| Baseline Zeros | 0 | 0 | 0 | 0 | 0 | 0 | 0 | 0 | 0 |
| Baseline Average | 0 | 0 | 0 | 0 | 0 | 0 | 0 | 0 | 0 |
| Reservoir [30, 51, 57] | 306 | 424 | 637 | 185 | 107 | 28 | 26 | 64 | 245 |
| KAN [46] | 1367 | 77 | 1797 | 159 | 1495 | 2406 | 1851 | 2286 | 1840 |
| HigherOrder DMD [37] | 2 | 4 | 2 | 3 | 2 | 0 | 1 | 4 | 5 |
| OptDMD [4] | 78 | 77 | 89 | 57 | 46 | 1 | 1 | 11 | 15 |
| PyKoopman [7, 56] | 44 | 2 | 45 | 3 | 62 | 1 | 0 | 16 | 3 |
| LSTM [24] | 3243 | 369 | 1414 | 835 | 728 | 50 | 50 | 1830 | 1171 |
| ODE-LSTM [15] | 22067 | 2270 | 2506 | 21957 | 17956 | 375 | 282 | 17238 | 1535 |
| Spacetime [68] | 6611 | 13981 | 1952 | 9439 | 6715 | 19 | 22 | 1110 | 3280 |
| DeepONet [47] | 1348 | 118 | 2414 | 334 | 2817 | 160 | 36 | 1965 | 6272 |
| SINDy [8, 19] | 53950 | 157 | 9 | 24 | 6731 | 139 | 649 | 16 | 348 |
| FNO [40] | 762 | 930 | 2154 | 597 | 2877 | 17 | 10 | 2852 | 30 |
| NeuralODE [12] | 2841 | 1635 | 421 | 451 | 196 | 39 | 21.24 | 4528 | 2957.52 |

(b) Mean Wall Clock Times on Kuramoto–Sivashinsky Dataset in Seconds

## 3 Foundation Model Results

We evaluated the performance of several widely used foundation models on our CTF. Each of these models is advertised as being capable of performing zero-shot time-series forecasting. The results are presented in Table 5. As the foundation models are pre-trained, we did not perform hyperparameter tuning or training. Instead, we provide their one-shot results, reflecting how such models would typically be used in real-world applications.

Table 5: Foundation model performances for each metric on each dataset

| Model | avg_score | E1 | E2 | E3 | E4 | E5 | E6 | E7 | E8 | E9 | E10 | E11 | E12 |
|---|---|---|---|---|---|---|---|---|---|---|---|---|---|
| Sundial [44] | 45.26 | **53.24** | **40.30** | 50.94 | **39.68** | 45.32 | **34.94** | 45.19 | 42.04 | 52.19 | **44.95** | 47.37 | 47.01 |
| TabPFN [26] | 28.80 | 51.35 | -26.27 | **84.06** | -26.80 | **79.02** | -14.27 | 31.49 | **58.00** | 28.85 | 27.60 | 22.54 | 29.96 |
| Panda [36] | -59.60 | -69.13 | -38.51 | -100.00 | -38.21 | -100.00 | -41.20 | -97.19 | -36.21 | -51.01 | -35.09 | -56.99 | -51.60 |
| Chronos [3] | -7.27 | 34.80 | -84.67 | 52.85 | -86.53 | 53.40 | -88.00 | 44.18 | -88.47 | **54.01** | -85.13 | 49.24 | **57.04** |
| Moirai [43] | -12.07 | 49.96 | -88.53 | 29.74 | -84.33 | 25.61 | -84.67 | **55.25** | -87.20 | 52.28 | -90.73 | **50.06** | 27.75 |
| LLMTime [20] | -36.89 | 4.59 | -91.40 | 0.59 | -100.00 | 0.44 | -94.47 | 4.34 | -93.73 | 4.10 | -94.47 | 8.38 | 8.99 |

(a) Model Scores on Lorenz Dataset

| Model | avg_score | E1 | E2 | E3 | E4 | E5 | E6 | E7 | E8 | E9 | E10 | E11 | E12 |
|---|---|---|---|---|---|---|---|---|---|---|---|---|---|
| Sundial [44] | 1.13 | 89.40 | **5.42** | **-4.60** | 1.73 | **-1.06** | 0.38 | 6.95 | 12.67 | -28.22 | **26.72** | -54.62 | -41.15 |
| TabPFN [26] | -2.51 | **97.91** | 3.65 | -100.00 | **2.01** | -100.00 | 1.17 | 3.66 | **30.91** | -32.50 | 24.74 | **12.67** | **25.70** |
| Chronos [3] | -15.06 | 97.48 | 4.08 | -100.00 | 1.81 | -100.00 | **7.75** | 13.93 | 2.58 | **-17.58** | 4.97 | -54.62 | -41.15 |
| Moirai [43] | -53.05 | 93.69 | -100.00 | -25.53 | -100.00 | -100.00 | -100.00 | **22.98** | -100.00 | -32.01 | -100.00 | -54.62 | -41.15 |
| Panda [36] | -89.23 | -5.84 | -100.00 | -100.00 | -100.00 | -100.00 | -100.00 | -100.00 | -100.00 | -100.00 | -100.00 | -100.00 | -64.89 |
| LLMTime [20] | -100.00 | -100.00 | -100.00 | -100.00 | -100.00 | -100.00 | -100.00 | -100.00 | -100.00 | -100.00 | -100.00 | -100.00 | -100.00 |

(b) Model Scores on Kuramoto–Sivashinsky Dataset

# 4 Models

## 4.1 Baselines

We implement two baseline models. One of the baselines predicts all zeros. The other baseline predicts the average of the input data per spatial dimension. We do not perform hyperparameter optimization for either of these models.

## 4.2 LSTM/ODE-LSTM

LSTM networks are a specialized type of recurrent neural network (RNN) designed to address the vanishing gradient problem inherent in traditional RNNs [24]. They achieve this through a unique architecture featuring memory cells and gating mechanisms (input, forget, and output gates), which regulate the flow of information over time. These gates enable LSTMs to selectively retain or discard historical data, making them particularly adept at capturing long-term dependencies in sequential data. In time-series forecasting, LSTMs excel at modeling temporal patterns, such as trends, seasonality, and irregular fluctuations, by leveraging past observations to predict future values. Their ability to handle complex, non-linear relationships and variable-length input sequences makes them a robust choice for tasks like stock prediction, energy load forecasting, or weather modeling, where historical context is critical to accurate predictions.

ODE-LSTMs are a flavor of LSTMs that try to further tackle the vanishing gradient problem by using an ODE solver to model the hidden state of the LSTM [15]. They show that traditional LSTMs can still suffer from a vanishing or exploding gradient and provide theory demonstrating ODE-LSTMs do not suffer from either of these problems.

We evaluate both a classical LSTM and the ODE-LSTM by searching over the following hyperparameters: hidden_state_size (dimension of the latent space), seq_length (input sequence length), and lr (learning rate).

| hyperparameter | type | min (or options) | max (or none) |
|---|---|---|---|
| hidden_state_size | randint | 3 | 32 |
| seq_length | randint | 5 | 512 |
| lr | log_uniform | $10^{-5}$ | $10^{-2}$ |

Table 6: Hyperparameter search space for the ODE-LSTM and LSTM models on metrics $E_1$ through $E_6$ for Lorenz. We train with a batch size of 128 for 200 epochs.

| hyperparameter | type | min (or options) | max (or none) |
|---|---|---|---|
| hidden_state_size | randint | 8 | 256 |
| seq_length | randint | 5 | 512 |
| lr | log_uniform | $10^{-5}$ | $10^{-2}$ |

Table 7: Hyperparameter search space for the ODE-LSTM and LSTM models on metrics $E_1$ through $E_6$ for Kuramoto-Sivashinsky. We train with a batch size of 128 for 200 epochs.

| hyperparameter | type | min (or options) | max (or none) |
|---|---|---|---|
| hidden_state_size | randint | 3 | 32 |
| seq_length | randint | 5 | 74 |
| lr | log_uniform | $10^{-5}$ | $10^{-2}$ |

Table 8: Hyperparameter search space for the ODE-LSTM and LSTM models on metrics $E_7$ through $E_{12}$ for Lorenz. We train with a batch size of 5 for $E_7$ through $E_{10}$ and a batch size of 128 for $E_{11}$ and $E_{12}$ for 200 epochs.

| hyperparameter | type | min (or options) | max (or none) |
|---|---|---|---|
| hidden_state_size | randint | 8 | 256 |
| seq_length | randint | 5 | 74 |
| lr | log_uniform | $10^{-5}$ | $10^{-2}$ |

Table 9: Hyperparameter search space for the ODE-LSTM and LSTM models on metrics $E_7$ through $E_{12}$ for Kuramoto-Sivashinsky. We train with a batch size of 5 for $E_7$ through $E_{10}$ and a batch size of 128 for $E_{11}$ and $E_{12}$ for 200 epochs.

## 4.3 SpaceTime

State-Space Models (SSMs) are mathematical frameworks that describe systems using latent (hidden) states evolving over time, observed through measurable outputs. They are widely used in control theory, signal processing, and time-series analysis to model dynamic systems. Modern adaptations like S4 (Structured State Space for Sequence Modeling) and SpaceTime are deep learning variants of SSMs tailored for sequential data. These models parameterize state transitions with structured matrices to efficiently capture long-range dependencies while remaining computationally tractable. Unlike LSTMs, SSMs are particularly effective at time-series forecasting of long-range dependencies with minimal memory overhead.

SpaceTime [68] is one such SSM that claims to be a state-of-the-art model on time-series forecasting and classification tasks. The authors claim that their model captures "complex, long-range, and *autoregressive*" dependencies, can forecast over long horizons, and is efficient during training and inference. They demonstrate improved performance over the popular S4 SSM and NLinear.

Based on the hyperparameter optimization described in the original paper and the hyperparameters which can be adjusted in the publicly available code, we do a hyperparameter search over the following values: lag (input sequence length), horizon (output sequence length), n_blocks (number of SpaceTime layers in the model encoder), dropout, weight_decay, kernel_dim (dimension of SSM kernel in each block), and lr (learning rate).

| hyperparameter | type | min (or options) | max (or none) |
|---|---|---|---|
| lag | randint | 32 | 512 |
| horizon | randint | 32 | 512 |
| n_blocks | choice | {3,4,5,6} | . |
| dropout | choice | {0, 0.25} | . |
| weight_decay | choice | {0, 0.0001} | . |
| kernel_dim | choice | {32,64,128} | . |
| lr | log_uniform | $10^{-5}$ | $10^{-2}$ |

Table 10: Hyperparameter search space for the SpaceTime model on metrics $E_1$ through $E_6$ for Lorenz and Kuramoto-Sivashinsky. We train with a batch size of 128 for 200 epochs.

| hyperparameter | type | min (or options) | max (or none) |
|---|---|---|---|
| lag | randint | 10 | 45 |
| horizon | randint | 10 | 45 |
| n_blocks | choice | {3,4,5,6} | . |
| dropout | choice | {0, 0.25} | . |
| weight_decay | choice | {0, 0.0001} | . |
| kernel_dim | choice | {32,64,128} | . |
| lr | log_uniform | $10^{-5}$ | $10^{-2}$ |

Table 11: Hyperparameter search space for the SpaceTime model on metrics $E_7$ through $E_{10}$ for Lorenz and Kuramoto-Sivashinsky. We train with a batch size of 5 for 200 epochs.

| hyperparameter | type | min (or options) | max (or none) |
|---|---|---|---|
| lag | randint | 10 | 45 |
| horizon | randint | 10 | 45 |
| n_blocks | choice | {3,4,5,6} | . |
| dropout | choice | {0, 0.25} | . |
| weight_decay | choice | {0, 0.0001} | . |
| kernel_dim | choice | {32,64,128} | . |
| lr | log_uniform | $10^{-5}$ | $10^{-2}$ |

Table 12: Hyperparameter search space for the SpaceTime model on metrics $E_{11}$ through $E_{12}$ for Lorenz and Kuramoto-Sivashinsky. We train with a batch size of 128 for 200 epochs.

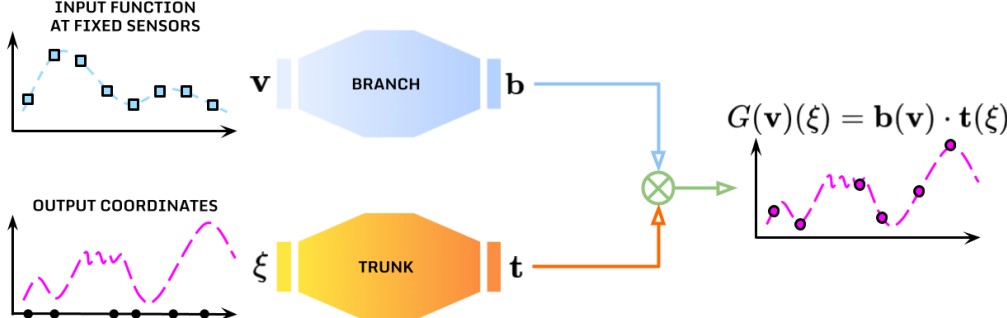

Figure 1: Architecture of the Deep Operator Network. The target field at the evaluation point $\xi$ is approximated by the inner product of the outputs of the branch net, which takes as input the measurements $\mathbf{v}$ of the input function $v \in \mathcal{V}$ and returns a set of coefficients $\mathbf{b}(\mathbf{v})$, and the trunk net, which encodes the coordinates $\xi$ into a vector $\mathbf{t}(\xi)$.

## 4.4 Deep Operator Networks

Deep Operator Networks (DeepONets) [47] recently emerged as a powerful tool designed to efficiently model high-dimensional physical systems and complex input-output relationships, as well as to solve challenging problems in scientific machine learning and engineering, such as partial differential equations. Specifically, DeepONets are a class of neural operators which decompose an operator $G : \mathcal{V} \rightarrow \mathcal{U}$ between infinite-dimensional functional spaces $\mathcal{V}$ and $\mathcal{U}$ into two cooperating sub-networks, namely *branch* and *trunk net*. The trunk encodes the input function $v \in \mathcal{V} : \Omega' \subset \mathbb{R}^d \rightarrow \mathbb{R}^{n_v}$ – which is typically sampled at a finite set of $n$ fixed sensors, resulting in the measurement vector $\mathbf{v} \in \mathbb{R}^{n \cdot n_v}$ – into $p$ coefficients $\mathbf{b}(v) \in \mathbb{R}^p$. Instead, the branch net provides the evaluation of a neural learnable $p$-dimensional basis $\mathbf{t}(\xi) \in \mathbb{R}^p$ at the spatial coordinates $\xi$ in the domain $\Omega \subset \mathbb{R}^d$. Doing so, the value of the output function $u \in \mathcal{U} : \Omega \rightarrow \mathbb{R}^{n_u}$ at the evaluation point $\xi \in \Omega$ is approximated through the basis expansion

$$u(\xi) = G(v)(\xi) \approx \mathbf{b}(v) \cdot \mathbf{t}(\xi).$$

See [47, 13, 48] for a complete presentation of DeepONets, including also universal approximation theorems for operators. A graphical summary of the DeepONet architecture is available in Figure 1.

**DeepONets for dynamical systems** DeepONets are versatile neural architectures designed to learn mappings between functional spaces. DeepONets are traditionally exploited for inferring the space-time evolution of physical variables, such as the solution of partial differential equations, starting from known quantities, such as forcing terms, initial conditions, parameters or control variables [47, 48, 66, 32]. However, it is possible to adapt and employ DeepONets in the proposed CTF in order to model and forecast time-series data and dynamical systems, as proposed by, e.g., [10, 11, 42, 23, 22, 54]. Specifically, we consider the operator

$$u_t(\xi) = G(u_{t-1}, ..., u_{t-k})(\xi) \approx \mathbf{b}(\mathbf{u}_{t-1}, ..., \mathbf{u}_{t-k}) \cdot \mathbf{t}(\xi)$$

where $u_t : \Omega \to \mathbb{R}^{n_u}$ and $\mathbf{u}_t \in \mathbb{R}^n$ are, respectively, the solution of the dynamical system under investigation at time $t$ and the corresponding spatial discretization, $k$ is the lag parameter and $\xi \in \Omega \subset \mathbb{R}^d$ are the spatial coordinates where to predict the evolution of the dynamics. Along with the evaluation point $\xi$, the trunk input may be enlarged with the time instance $t$ or the time-step $\Delta t$, as proposed by [48, 42].

**DeepONets implementation**    The implementation of DeepONets within the proposed CTF is based on the *DeepXDE* library [49]. In particular, when dealing with forecasting tasks, we predict the state evolution in an autoregressive manner, and we enlarge the trunk input with the time-step $\Delta t$, as it results in better performance. As proposed by [48], we consider a scaler to normalize the data before training. Moreover, we employ branch and trunk networks with the same number of neurons per hidden layer, so as to reduce the number of hyperparameters.

The Kuramoto-Sivashinsky dataset deals with one-dimensional scalar-valued functions, that is $d = n_u = 1$. The KS solution is discretized and evaluated at $n = 1024$ spatial points uniformly spaced in the domain $\Omega = [0, 32\pi]$. Notice that we take into account the same locations across all the input-output pairs, resulting in a lower computational cost.

The Lorenz test case, instead, considers a three-dimensional state variable evolving over time, without spatial dependence. Among different alternatives, we adapt DeepONet in this context by considering the fictitious domain $\Omega = 1, 2, 3$ and the state function $u_t : \Omega = \{1, 2, 3\} \to \mathbb{R}$ mapping the index $\xi \in \Omega = \{1, 2, 3\}$ into the $\xi$-th component of the state vector at time $t$. For instance, if $\xi = 1$, DeepONet predicts the evolution of the first component of the state variable starting from the past state values encoded by the branch net.

**Hyperparameters**    The DeepONet hyperparameters mainly concern the neural network architectures and the corresponding training procedure. In addition, the lag parameter determines the length of the past state history fed into the branch input for forecasting. Notice that the lag value cannot be larger than the dimension of burn-in data, and it is set equal to zero when dealing with reconstruction tasks. Table 13 provides a summary of the hyperparameters in play, along with the corresponding search spaces explored for hyperparameters tuning.

| hyperparameter | type | min (or options) | max (or none) |
|---|---|---|---|
| lag | integer | 1 | 99 |
| branch_layers | integer | 1 | 5 |
| trunk_layers | integer | 1 | 5 |
| neurons | integer | 1 | 512 |
| activation | choice | {"tanh", "relu", "elu"} | . |
| initialization | choice | {"Glorot normal", "He normal"} | . |
| optimizer | choice | { "adam", "L-BFGS" } | . |
| learning_rate | loguniform | $10^{-5}$ | $10^{-1}$ |
| epochs | integer | 10000 | 10000 |

Table 13: Hyperparameter search space for DeepONet.

## 4.5    Sparse Identification of Nonlinear Dynamics

Sparse Identification of Nonlinear Dynamics (SINDy) [8] is a powerful algorithm designed to discover interpretable and parsimonious governing equations from time-series data. Given the data matrices

$$X = \begin{bmatrix} x_1(t_1) & x_1(t_2) & ... & x_1(t_m) \\ \vdots & \vdots & \ddots & \vdots \\ x_n(t_1) & x_n(t_2) & ... & x_n(t_m) \end{bmatrix} ; \quad \dot{X} = \begin{bmatrix} \dot{x}_1(t_1) & \dot{x}_1(t_2) & ... & \dot{x}_1(t_m) \\ \vdots & \vdots & \ddots & \vdots \\ \dot{x}_n(t_1) & \dot{x}_n(t_2) & ... & \dot{x}_n(t_m) \end{bmatrix}$$

collecting, respectively, the state vector $\mathbf{x}(t) = [x_1(t), ..., x_n(t)]$ and the corresponding time derivatives $\dot{\mathbf{x}}(t) = [\dot{x}_1(t), ..., \dot{x}_n(t)]$ at the time instances $t_1, ..., t_m$, we aim at identifying the (possibly nonlinear) underlying governing equation $\dot{\mathbf{x}}(t) = f(\mathbf{x}(t))$. To this aim, SINDy considers the following approximation

$$\dot{X} = \Theta(X)\Xi$$

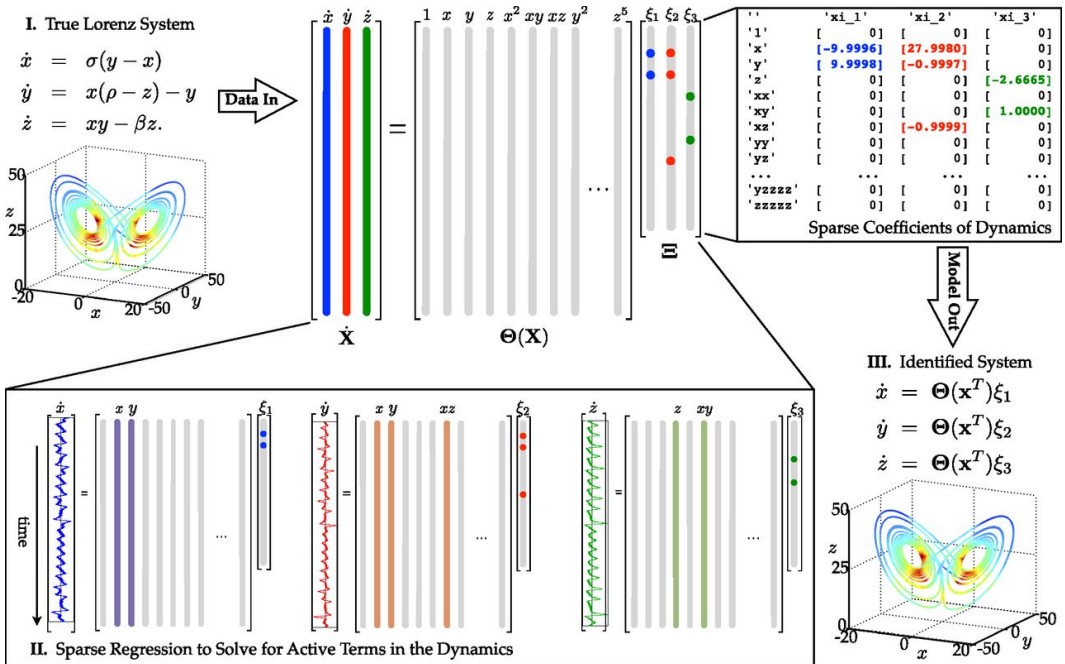

Figure 2: Schematic of the Sparse Identification of Nonlinear Dynamics (SINDy) algorithm from [8], demonstrated on the Lorenz equations. The temporal evolution of the state variable and its derivative are collected in the data matrices $X$ and $\dot{X}$. The dynamical system $\dot{X} = \Theta(X)\Xi$ is then identified through sparsity promoting algorithms.

where $\Theta(X)$ is a library of candidate regression terms, such as polynomials or trigonometric functions, while $\Xi$ are the corresponding regression coefficients. Sparsity promoting strategies are crucial to identify simple and interpretable dynamical systems, capable of avoiding overfitting and accurately extrapolating beyond training data. In particular, the regression coefficients $\Xi$ are determined through sparse regression strategies, such as Least Absolute Shrinkage and Selection Operator (LASSO) or Sequentially Thresholded Least SQuares (STLSQ). See Figure 2 for a scheme of the SINDy algorithm on the Lorenz system.

SINDy can easily handle parametric dependencies: indeed, augmenting the state vector with the (possibly time-dependent) parameter values $\boldsymbol{\mu}$ and adding $\boldsymbol{\mu}$-dependent terms in the library $\Theta(X, \boldsymbol{\mu})$, it is possible to identify parametric sparse dynamical systems.

Identifying sparse dynamical systems from high-dimensional data may be computationally expensive. A possible workaround is given by dimensionality reduction techniques, such as Proper Orthogonal Decomposition (POD) [8] or autoencoders [9], which project state snapshots onto a low-dimensional manifold. SINDy can thus be applied on the low-dimensional latent variables, allowing for efficient and accurate forecasting of the high-dimensional state evolution.

**SINDy implementation**  The implementation of SINDy is based on the *PySINDy* library [16]. After collecting the data and approximating the time derivatives through numerical schemes, the SINDy algorithm is applied to identify a sparse dynamical system describing the data evolution over time. The integrator *solve_ivp* by *scipy* [64] is considered to simulate the system and to predict future state values. Notice that, whenever the identified model is very complex and the integrator fails, the static dynamical system $\dot{\mathbf{x}} = 0$ is employed.

The Kuramoto-Sivashinsky dataset deals with the temporal evolution of a chaotic partial differential equation on the spatial domain $[0, 32\pi]$. The KS solution is discretized and evaluated at $n = 1024$ locations, resulting in a collection of high-dimensional snapshots over time. Proper Orthogonal Decomposition (POD) is thus exploited to compress the temporal data, and SINDy is applied to identify the dynamics of the most energetic POD coefficients. Therefore, the KS predictions are

retrieved by integrating the SINDy model and projecting the POD coefficients onto the original high-dimensional state space.

Parametric SINDy models are considered when testing the ability of the model to generalize to different parameter values. Since the parameter values employed for data generation are not publicly available, we take into account fictitious values mimicking the interpolatory and extrapolatory regimes.

**Hyperparameters** The SINDy algorithm can exploit different differentiation methods to approximate time derivatives, different terms in the library $\Theta(X)$ – such as, e.g., polynomials and/or trigonometric functions up to a chosen order – as well as different sparse regression techniques. Table 14 provides a summary of the hyperparameters in play, along with the corresponding search spaces explored for hyperparameter tuning.

| hyperparameter | type | min (or options) | max (or none) |
|:---:|:---:|:---:|:---:|
| POD_modes | integer | 1 | 50 |
| differentiation_method | choice | { "finite_difference", "spline", "savitzky_golay", "spectral", "trend_filtered", "kalman" } | . |
| differentiation_method_order | integer | 1 | 10 |
| feature_library | choice | { "polynomial", "Fourier", "mixed" } | . |
| feature_library_order | integer | 1 | 10 |
| optimizer | choice | {"STLSQ", "SR3", "SSR", "FROLS"} | . |
| threshold | choice | { "adam", "L-BFGS" } | . |
| learning_rate | loguniform | $10^{-3}$ | $10^3$ |
| alpha | loguniform | $10^{-3}$ | $10^1$ |

Table 14: Hyperparameter search space for SINDy. The POD_modes parameter has an effect only for the Kuramoto-Sivashinsky test case.

## 4.6 Dynamic Mode Decomposition

The Dynamic Mode Decomposition (DMD) is a data-driven method developed by Schmid [62] in the fluid dynamics community to identify spatio-temporal coherent structures from high-dimensional data. The DMD algorithm is based on the Singular Value Decomposition (SVD) of a data matrix; in particular, DMD is able to provide a modal decomposition where each mode consists of spatially correlated structures that have the same linear behaviour in time. The DMD method is found to have a significant connection with the Koopman operator theory [60]: in particular, the DMD can be formulated as an algorithm able to learn the best-fit linear dynamical system to advance in time (Figure 3).

There are many variants of DMD, connected to existing techniques from system identification and modal extraction [6]. Here, we will provide a brief overview of the underlying idea of the original DMD algorithm, from which all the other variants can be derived. The first step is to collect a set of snapshots of the system at different time steps. The data matrix is then constructed by stacking the snapshots in columns, i.e., $\mathbf{X} = [\mathbf{x}_1, \mathbf{x}_2, \ldots, \mathbf{x}_{N_t}] \in \mathbb{C}^{\mathcal{N}_h \times N_t}$, where $\mathbf{x}_k \in \mathbb{C}^{\mathcal{N}_h}$ is the $k$-th snapshot at time $t_k$ and $N_t$ is the number of snapshots. The original formulation from [62, 60] supposed uniform sampling in time, i.e. $t_k = k\Delta t$, where $\Delta t$ is the time step and $t_{k+1} = t_k + \Delta t$. Overall, the DMD algorithm seeks the leading spectral decomposition of the best-fit linear operator $\mathbb{A} \in \mathbb{C}^{\mathcal{N}_h \times \mathcal{N}_h}$ that advances the system in time, i.e.

$$\mathbf{x}_{k+1} \approx \mathbb{A}\mathbf{x}_k \quad \longleftrightarrow \quad \mathbf{X}_{[2:N_t]} \approx \mathbb{A}\mathbf{X}_{[1:N_t-1]}$$

As we said above, the DMD algorithm is based on the SVD of the data matrix $\mathbf{X}$ of rank $r$, which can be written as $\mathbb{X} \simeq \mathbf{U}\boldsymbol{\Sigma}\mathbf{V}^*$: $\mathbf{U} \in \mathbb{C}^{\mathcal{N}_h \times r}$ represents the left singular vectors and are also known as modes, describing the dominant spatial structures extracted from the data; the diagonal matrix $\boldsymbol{\Sigma} \in \mathbb{R}^{r \times r}$ contains the singular values, which are related to the energy/information retained by the modes; in the end, $\mathbf{V}^* \in \mathbb{C}^{r \times N_t}$ represents the right singular vectors, which are related to the

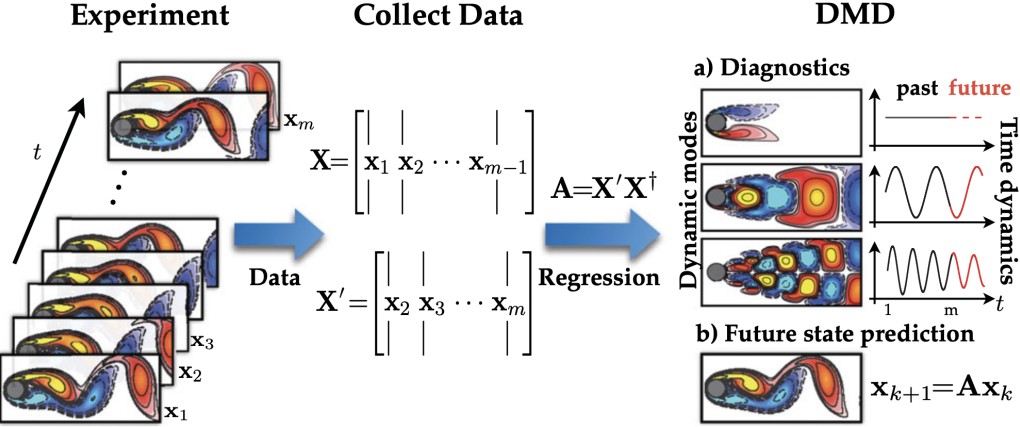

Figure 3: Scheme of the Dynamic Mode Decomposition algorithm from [28]. The data matrix $\mathbf{X}$ is constructed by stacking the snapshots in columns. The SVD of the data matrix is computed, and the dynamical matrix is fitted to the data. This allows us to compute the state of the system for future time instances.

temporal dynamics of the modes. This compression operation allows to compute the dynamical matrix $\mathbb{A}$ in a more efficient way [28, 6], avoiding the direct inversion of the high-dimensional snapshot matrix.

Indeed, in the literature different variants of DMD have been proposed: in this context, the High-Order DMD (HODMD) [38], which exploits time delay embedding to fit the optimal Koopman Operator, and the Optimised DMD (OptDMD) [5, 61], which is a variant of DMD that can also use the Bagging algorithm to improve the robustness of the DMD algorithm against noise. This latter variant has been shown to be the most robust and stable algorithm for real-world applications [18]. The implementation of the DMD algorithm is available in the `pyDMD` package [17, 27], which is a Python library for DMD and its variants. The library is designed to be easy to use and flexible, allowing users to customise the algorithm for their specific needs.

**Parametric DMD** The extension of DMD to parametric systems is a recent development in the field of system identification. Different approaches have been proposed in the literature; in this work, the implementation of Andreuzzi et al. [1] within `pyDMD` is adopted. Up to now, the package does not support the OptDMD algorithm directly, we have implemented a wrapper to use the OptDMD algorithm with the parametric DMD following the same approach of the package, based on the interpolation of the forecasted reduced dynamics. We appreciate that further work and rigorous testing of this implementation are planned for future work. Similar to SINDy, since the parameter values employed for data generation are not publicly available, fictitious values mimicking the interpolatory and extrapolatory regimes have been used.

**Hyperparameter tuning** The hyperparameters of the DMD algorithm depend on the specific variant adopted. Every DMD algorithm has a set of hyperparameters that can be tuned to improve the performance of the algorithm; however, the rank of the SVD is common to all of them and plays a crucial role in the reduction process. The HODMD algorithm also includes the delay embedding, defining the size of the lagging window to use. The OptDMD algorithm can also put constraints on the DMD eigenvalues to ensure that the dynamics follow a certain behaviour. In the end, the parametric DMD can operate in two different modes: partitioned and monolithic. The hyperparameters of both DMD algorithms are listed in Tables 15 and 16.

### 4.7 Koopman operator-based dynamic system prediction

**The Koopman operator** Koopman operator theory is a useful tool that has found increasing attention in the data-driven scientific computing community and can essentially be seen as an

| hyperparameter | type | min (or options) | max (or none) |
|---|---|---|---|
| rank | randint | 3 | 50 |
| delay | randint | 0 | 200 |
| parametric | choice | {"partitioned", "monolithic"} | |

Table 15: Hyperparameter search space for the HODMD algorithm for Lorenz and Kuramoto-Sivashinsky (the parametric hyperparameter has an effect only for metrics $E_{11}$ and $E_{12}$).

| hyperparameter | type | min (or options) | max (or none) |
|---|---|---|---|
| rank | randint | 3 | 50 |
| delay | randint | 0 | 100 |
| parametric | choice | { "partitioned", "monolithic"} | |
| eig_constraints | choice | { "none", "stable", "conjugate_pairs"} | |

Table 16: Hyperparameter search space for the OptDMD algorithm for Lorenz and Kuramoto-Sivashinsky (the parametric hyperparameter has an effect only for metrics $E_{11}$ and $E_{12}$).

extension of dynamic mode decomposition - viewing the statespace of the dynamic system through the lens of nonlinear observables. This point-of-view dates back to early work by [33, 34] and a modern review can be found in [7]. We outline the method briefly before describing the set-up for the chosen implementation and our testing on the CTF. Consider a dynamical system (either an ODE or a semi-discretisiation of a PDE) of the form:

$$\frac{d\mathbf{x}}{dt} = \mathbf{f}(\mathbf{x}),$$

where $\mathbf{f} : \mathbb{R}^N \to \mathbb{R}^N$ may be a nonlinear forcing. The central idea in Koopman operator theory is then to learn a coordinate transform (i.e. a set of nonlinear observables) $\Phi : \mathbb{R}^N \to \mathbb{R}^M$, under which the dynamics becomes (approximately) linear, i.e.

$$\frac{d\mathbf{z}}{dt} \approx \mathbf{A}\mathbf{z}, \quad \mathbf{z}(t) = \Phi(\mathbf{x}(t)).$$

In this new coordinate system, the exact solution of the linear dynamics is straightforward. The inference of $\Phi$ and $\mathbf{A}$ can be formulated as a regression problem.

**Numerical implementation and parameter choices**  In our current CTF test we use the `PyKoopman` Python library as the main reference point for the Koopman method for dynamic system prediction [56]. The Python package serves as a good reference since it is regularly maintained and has an up-to-date implementation of several central features of the Koopman operator framework. As mentioned above there are two central parameters that affect the performance of the Koopman method: the observables and the regression method. Exploiting the existing implementation in `PyKoopman` we allowed in our CTF testing the variation of the following set of parameters:

- Type of observable: Options include the identity, polynomials of variable degree, time delay (of variable depth), radial basis functions (of variable number) and random Fourier features, as well as the concatenation of all of the aforementioned observables with the identity;
- Type of regressor: DMD, EDMD, HAVOK and KDMD;
- Regressor rank;
- Least-squares regularisation and rank of the regularisation (this option is implemented only in EDMD and KDMD).

Note that in principle a neural network-based DMD is also implemented in the PyKoopman package, but in our fine-tuning we found that this lead consistently to worse performance than the above four types of regressors thus we did exclude it from the hyperparameter tuning.

**Parametric PyKoopman**  Out-of-the-box `PyKoopman` does not have a parametric implementation, thus in order to test the Koopman method on task 4, we loosely follow [2, 21] and implement a custom parametric version of `PyKoopman` by spline interpolation of the learned Koopman operator

and corresponding eigenfunctions. We acknowledge that further work and rigorous testing of various parametric versions of the Koopman method are required to identify the best performing implementation for task 4.

**Further comments on the use with chaotic systems**   We note that the performance of the Koopman operator on the KS and Lorenz system is notably subpar, especially when compared to results reported in prior work [55]. This is not unexpected and a likely source of challenge is the chaotic nature of both equations, which has also been noticed by the authors of the `PyKoopman` package. Essentially, in chaotic systems there may not be a dominating low-rank structure that can be learned and exploited with the Koopman method (cf. the section on "Unsuccessful examples of using Dynamic mode decomposition on PDE system" in [55]).

**Hyperparameter tuning**   Based on the available choices implemented in the PyKoopman package and the examples described in the documentation [55], we performed a hyperparameter search over the following parameters: type of observable and potential concatenation with the identity, observables integer parameter (representing the polynomial degree in case of polynomial observables, the number of time delay steps in the case of time delay observables and the parameter $D$ in the random Fourier feature case), the number of centers for the radial basis function observables, observables float parameter (representing the radial basis function kernel width and the parameter $\gamma$ in the radial basis function case respectively), regressor type, regressor rank, TLSQ rank (the regularisation rank called only when the regressor is EDMD and KDMD). The details of the parameter space explored are shown in Table 17.

| hyperparameter | type | min (or options) | max (or none) |
|---|---|---|---|
| observables | choice | {Identity, Polynomial, TimeDelay, RadialBasisFunctions, RandomFourierFeatures} | . |
| Identity concatenation | choice | {true, false} | |
| Integer parameter | randint | 1 | 10 |
| # RBF centers | randint | 10 | 1000 |
| Float parameter | uniform | 0.5 | 2.0 |
| regressor type | choice | {DMD,EDMD, HAVOK, KDMD} | . |
| regressor rank | randint | 1 | 200 |
| TLSQ rank | randint | 1 | 200 |

Table 17: Hyperparameter search space for the PyKoopman model.

## 4.8   Reservoir Computing

In its broadest sense, reservoir computing (RC) is a general machine learning framework for processing sequential data. RC functions by projecting data into a high-dimensional dynamical system and training a simple readout from these dynamics back to a quantity or signal of interest. Although there exists a large and ever-growing body of literature on leveraging physical systems to act as high-dimensional "reservoirs" [63], the most common form of RC remains an echo state network (ESN) [29, 52]. ESNs are a form of recurrent neural network (RNN) that have been demonstrated to achieve state-of-the-art performance in the forecasting of chaotic dynamical systems [58, 65]. We now introduce the specific form of ESN we use in evaluating performance on the CTF datasets, following many of the conventions presented in [58].

**ESNs for Lorenz63 system.**   Given a time series $u_0, \dots, u_T$, a randomly instantiated, high-dimensional dynamical system is evolved according to

$$h_{t+1} = (1 - \alpha)h_t + \alpha \tanh\left(W_{hh}h_t + W_{hu}u_t + \sigma_b \mathbf{1}\right) \tag{1}$$

where $\alpha$ is the so-called leak rate hyperparameter, $W_{hh}$ and $W_{hu}$ are fixed, random matrices, $\sigma_b$ is a bias hyperparameter and $\mathbf{1}$ denotes a vector of ones. $W_{hh} \in \mathbb{R}^{N_h \times N_h}$ ($N_h$ denotes the number of entries in $h$) is taken to be a random, sparse matrix with density $\approx 2\%$ and non-zero entries sampled from $\mathcal{U}(-1, 1)$ and then scaled such that the spectral radius of $W_{hh}$ is $\rho$. $W_{hu} \in \mathbb{R}^{N_h \times N_u}$ ($N_u$ denotes the number of entries in $u$) is a random matrix with each entry drawn independently from

$\mathcal{U}(-\sigma, \sigma)$. Initializing $h_0$ as $h_0 = \mathbf{0}$, we generate a sequence of training reservoir states $h_0, \ldots, h_T$. We discard the initial $N_{spin}$ training states as an initial transient and then perform a Ridge regression (with Tikhonov regularization $\beta$) to learn a linear map $W_{uh}$ such that

$$W_{uh} g(h_i) \approx u_i. \tag{2}$$

$g : N_h \rightarrow N_h$ is often taken to be the identity map or simply squaring every odd indexed entry of $h_i$. We assume the latter convention, following the work of Pathak et al [57]. Once trained the reservoir dynamics can be run autonomously as

$$h_{t+1} = (1 - \alpha)h_t + \alpha \tanh\left(W_{hh}h_t + W_{hu}W_{uh}g(h_t) + \sigma_b \mathbf{1}\right) \tag{3}$$

to obtain a forecast of arbitrary length. A summary of tunable hyperparameters for this architecture applied to the Lorenz system are presented in Table 18. $N_{spin} = 15$ for error metrics $E_7$ through $E_{10}$ and $N_{spin} = 100$ for all other metrics.

**ESNs for KS system.** RC approaches typically rely on the latent dimension $N_h >> N_u$. However, the computational cost of the previous algorithm scales roughly quadratically with $N_h$. Thus, while the above approach works well for relatively small systems, without modification it does not scale well to large states such as those encountered in PDE simulations. Pathak et al. introduced a parallel reservoir approach to address this issue by dividing a high-dimensional input into $g$ lower dimensional "chunks" [57]. A single reservoir then accepts as input only $N_u/g + 2L$ values, where $L$ is a locality parameter that dictates the overlap of input for two adjacent reservoirs. The output of the single reservoir is only $g$ entries of the state. Since computational cost grows linearly in the number of reservoirs, this parallel approach allows for the application of RC to higher dimensional systems. Each individual reservoir is trained exactly as for the Lorenz system; there are now just $g$ reservoirs representing different regions of the domain.

Since we introduce two new hyperparameters in the parallel setup ($L$ and $g$), when we perform our hyperparameter tuning for the KS system we fix $\alpha = 1$ and $\sigma_b = 0$, following the work of Pathak et al. The complete hyperparameter search space for the KS system is given in Table 19.

| hyperparameter | type | min (or options) | max (or none) |
|---|---|---|---|
| $\alpha$ | uniform | 0 | 1 |
| $\sigma$ | loguniform | 0.0001 | 1.0 |
| $\sigma_b$ | uniform | 0 | 2 |
| $\rho$ | uniform | 0.02 | 1 |
| $\beta$ | loguniform | $10^{-10}$ | $10^{-1}$ |
| $N_h$ | randint | 500 | 3000 |

Table 18: Hyperparameter search space for the reservoir model on the Lorenz 63 system.

| hyperparameter | type | min (or options) | max (or none) |
|---|---|---|---|
| $g$ | choice | $\{16, 32, 64, 128\}$ | $\cdot$ |
| $\sigma$ | loguniform | 0.0001 | 1.0 |
| $L$ | randint | 1 | 10 |
| $\rho$ | uniform | 0.02 | 1 |
| $\beta$ | loguniform | $10^{-10}$ | $10^{-1}$ |
| $N_h$ | randint | 500 | 3000 |

Table 19: Hyperparameter search space for the reservoir model on the KS system.

## 4.9 Fourier Neural Operator

Neural operators are a class of machine learning models designed to learn mappings between function spaces, in contrast to the finite-dimensional Euclidean spaces typically used in conventional neural networks. Although the inputs and outputs are discretized in practice, neural operators aim to generalize across discretizations and treat functions as the primary objects of learning.

The Fourier Neural Operator (FNO), in particular, is a neural operator architecture that replaces the kernel integral operator with a convolution operator defined in Fourier space, which allows for

learning of operators in the frequency domain. It maps the input to the frequency domain using the Fourier transform, applies spectral convolution by multiplying learnable weights with the lower Fourier modes, and maps the result back to the physical domain via the inverse Fourier transform. This allows the model to learn families of PDEs, rather than solving individual instances. Without the high cost of evaluating integral operators, it maintains competitive computational efficiency.

Let $D \subset \mathbb{R}^d$ be a bounded domain. We consider learning an operator $G$ that maps between function spaces:

$$G : \mathcal{A} \to \mathcal{U} \tag{4}$$

where $\mathcal{A} = L^2(D; \mathbb{R}^{d_a})$ is the input function space and $\mathcal{U} = L^2(D; \mathbb{R}^{d_u})$ is the output function space.

Given an input function $a \in \mathcal{A}$, the FNO approximates the operator $G$ through a kernel integral operator:

$$G(a)(x) = \sigma \left( Wa(x) + b + \int_D \kappa(x, y) a(y) \, dy \right) \tag{5}$$

where $W \in \mathbb{R}^{d_u \times d_a}$ is a linear transformation, $b \in \mathbb{R}^{d_u}$ is a bias term, $\kappa : D \times D \to \mathbb{R}^{d_u \times d_a}$ is a learnable kernel function, and $\sigma : \mathbb{R}^{d_u} \to \mathbb{R}^{d_u}$ is a pointwise non-linear activation function.

The kernel is parameterized in Fourier space as:

$$\kappa(x, y) = \sum_{k \in \mathbb{Z}^d} \widehat{\kappa}(k) e^{2\pi i k \cdot (x-y)} \tag{6}$$

where $\widehat{\kappa}(k)$ are the Fourier coefficients of the kernel. The translation-invariant kernel $\kappa(x, y) = \kappa(x - y)$ enables efficient convolution. This leads to the implementation:

$$G(a)(x) = \sigma \left( Wa(x) + b + \sum_{k \in \mathbb{Z}^d} \widehat{\kappa}(k) \widehat{a}(k) e^{2\pi i k \cdot x} \right) \tag{7}$$

where $\widehat{a}(k)$ represent the Fourier coefficients of the input function $a$. In practice, the sum over $k \in \mathbb{Z}^d$ is truncated to a finite number of low-frequency modes.

**Model Architecture** The architecture (Figure 4) begins with an initial fully connected multilayer perceptron (MLP) that projects the input to a higher-dimensional space, followed by four Fourier layers, and concludes with two fully connected MLPs that project the output to the desired dimensions.

Each Fourier layer performs a spectral convolution by first transforming the data into the frequency domain using Fast Fourier Transform (FFT), then multiplying the Fourier coefficients with learable weights in the frequency space, and finally transforming back to physical space using inverse FFT. The Fourier layer only keeps a limited number of the lower Fourier modes, with high modes being filtered out. Additionally, each layer adds a linearly transformed version of its input to the output of the spectral convolution, which helps preserve local features and adds flexibility to the layer's expressiveness. Every Fourier layer is followed by a GELU activation function to introduce non-linearity.

**Hyperparameters** Based on our implementation of the FNO model, which closely follows that of the original paper, we test the hyperparameters as shown in Table 20. The number of Fourier modes is tuned separately for each mode.

| hyperparameter | type | range or options |
|---|---|---|
| Fourier modes | integer | [8,32] |
| Network width | integer | [32, 128] |
| Batch size | choice | 16, 32, 64, 128 |
| Learning rate (lr) | loguniform | [0.0001, 0.01] |

Table 20: Hyperparameter search space for the FNO model.

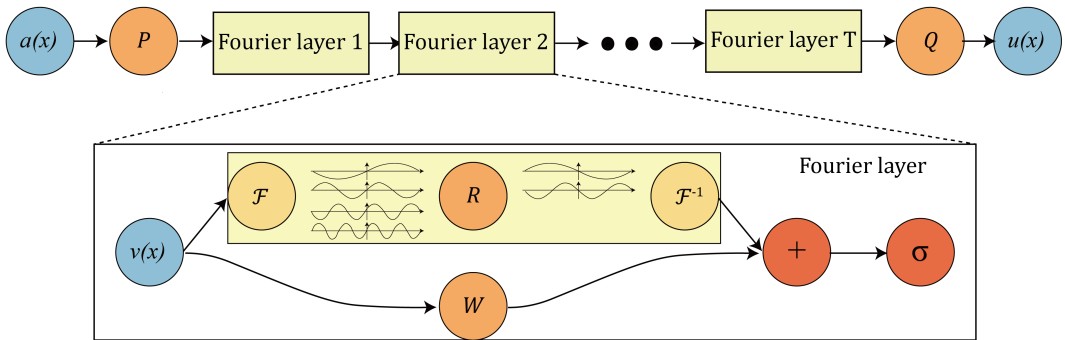

Figure 4: Architecture of the Fourier Neural Operator from [40]

## 4.10 Kolmogorov-Arnold Networks

Kolmogorov–Arnold Networks (KANs) are a recently proposed alternative to traditional Multi-Layer Perceptrons (MLPs) [45]. With learnable activation functions placed on edges that replace linear weights, KANs have been shown to provide improved accuracy and greater interpretability compared to traditional methods.
KANs were inspired by the Kolmogrov-Arnold representation theorem which posits that any multivariate continuous function $f$ on a bounded domain can be expressed as a finite composition and addition of univariate continuous functions [31]. In other words, for a smooth function $f : [0,1]^n \to \mathbb{R}$,

$$f(\mathbf{x}) = f(x_1, x_2, ..., x_n) = \sum_{q=1}^{2n+1} \Phi_q \left( \sum_{p=1}^{n} \phi_{q,p}(x_p) \right) \tag{8}$$

where $\phi_{q,p} : [0,1] \to \mathbb{R}$ and $\Phi_q : \mathbb{R} \to \mathbb{R}$.

**Model Architecture**    While the Kolmogrov-Arnold representation theorem is restricted to a small number of terms and only two hidden layers, this theorem can be generalized to increase the width and depth of the network. A single KAN layer is defined as a matrix of 1D functions thus the inner and outer functions in Equation 8, $\phi_{q,p}$ and $\Phi_q$, each represent a single KAN layer. A deeper network can be constructed by adding more KAN layers. A general KAN network with $L$ layers can be represented as a composition of $L$ functions:

$$f(\mathbf{x}) = \sum_{i_{L-1}=1}^{n_{L-1}} \phi_{L-1,i_L,i_{L-1}} \left( \sum_{i_{L-2}=1}^{n_{L-2}} \cdots \left( \sum_{i_2=1}^{n_2} \phi_{2,i_3,i_2} \left( \sum_{i_1=1}^{n_1} \phi_{1,i_2,i_1} \left( \sum_{i_0=1}^{n_0} \phi_{0,i_1,i_0}(x_{i_0}) \right) \right) \right) \cdots \right)$$

where $n_l$ is the number of nodes in the $l^{th}$ layer and $\phi_{l,j,k}$ is the activation function that connects the $k^{th}$ neuron in the $l^{th}$ layer to the $j^{th}$ neuron in the $l+1$ layer. The network architecture is better illustrated in Figure 5 which was adapted from Figure 2.2 in [45].

Each activation function is comprised of a basis function $b(x)$ and a spline function:

$$\phi(x) = w_b b(x) + w_s \text{spline}(x)$$

where

$$b(x) = \text{silu}(x) = \frac{x}{1 + e^{-x}}$$

$$\text{spline}(x) = \sum_i c_i B_i(x)$$

Initially, $w_s$ is set to 1 and $\text{spline}(x) \approx 0$. The weights of the basis function is initialized according to Xavier initializations.

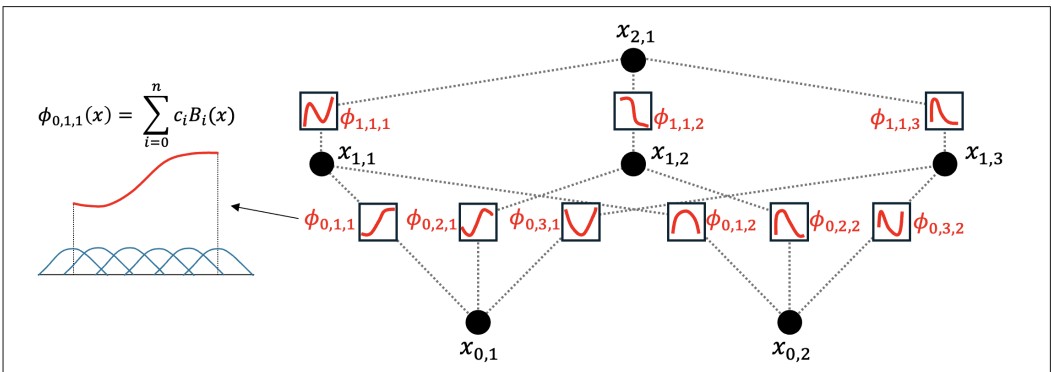

Figure 5: Sample architecture of a Kolmogorov-Arnold Network with three layers of size $[2, 3, 1]$. Activation functions $\phi$ are placed on the edges and are parametrized as a spline. Each output of a node is a sum of its inputs.

**KAN Implementation** Although KANs have primarily been applied to science-related tasks such as function approximation and PDE solving, Example 14 of the `pykan` package demonstrates their use in a supervised learning setting. In this work, the KAN implementation from that example was adapted to address the reconstruction and forecasting tasks posed in the Common Task Framework.

For forecasting tasks, the input-output pairs were constructed in an autoregressive manner, where each input consisted of lagged observations used to predict future values. The input and output dimensions depend on both the number of spatial dimensions in the dataset and the chosen lag. The Lorenz 63 system is a three-dimensional dynamical system. For a lag of $l$, the input dimension was set to $d_{\text{in}} = 3l$. While prediction windows greater than 1 were tested during training, a prediction window of 1 produced the best results. Therefore, the output dimension was fixed at $d_{\text{out}} = 3$. For the Kuramoto–Sivashinsky (KS) dataset, which contains 1024 spatial points, the input dimension was set to $d_{\text{in}} = 1024l$ and the output dimension to $d_{\text{out}} = 1024$.

For reconstruction tasks, the model was trained in an autoencoding fashion, where each input was mapped directly to itself as the target output. For the Lorenz 63 system, the input and output dimensions were both set to $d_{\text{in}} = d_{\text{out}} = 3$. For the Kuramoto–Sivashinsky (KS) system, the dimensions were set to $d_{\text{in}} = d_{\text{out}} = 1024$.

**Hyperparameters** Based on the hyperparameter settings provided in the `pykan` package and the results reported in the original paper [45], the hyperparameters outlined in Tables 21 and 22 were selected and tuned for this model. Broadly, the hyperparameters fall into two categories: (1) model architecture and (2) training.
Architecture-related hyperparameters include the number of layers, dimensions of hidden layers, grid resolution, the polynomial degree of the spline basis ($k$), and the lag. Training-related hyperparameters include the number of training steps (epochs), learning rate, overall regularization strength ($\lambda$), and the regularization coefficient for the spline parameters ($\lambda_{coef}$).

## 4.11 Physics-Informed Neural Networks

Physics-Informed Neural Networks (PINNs), introduced by Raissi et al. [59], have emerged as a powerful framework for solving differential equations using deep learning. Unlike standard neural networks, PINNs embed physical laws directly into the loss function, enabling them to honor both data fidelity and governing equations. The loss function is typically composed of two terms:

$$\mathcal{L}(\theta, \gamma) = \mathcal{L}_{\text{data}}(\theta) + \lambda \mathcal{L}_{\text{DE}}(\theta, \gamma) = \frac{1}{N_d} \sum_{i=1}^{N_d} \|u_\theta(x_i, t_i) - u(x_i, t_i)\|_2^2 + \lambda \frac{1}{N_f} \sum_{i=1}^{N_f} \|\mathcal{N}_\gamma[u_\theta(x_i, t_i)]\|_2^2,$$

| hyperparameter | type | min (or options) | max (or none) |
| --- | --- | --- | --- |
| steps | randint | 50 | $10^4$ |
| lag* | randint | 1 | 5 |
| lr | loguniform | $10^{-5}$ | $10^{-1}$ |
| num_layers | randint | 1 | 5 |
| {one-five}_dim** | randint | 1 | 10 |
| grid | randint | 1 | 100 |
| k | randint | 1 | 3 |
| $\lambda$ | loguniform | $10^{-7}$ | $10^{-3}$ |
| $\lambda_{coef}$ | loguniform | $10^{-7}$ | $10^{-3}$ |

Table 21: Hyperparameter search space for the KAN model on the Lorenz 63 system. NOTE: The lag parameter is set to zero for reconstruction tasks (pair_id = 2 or 4)*. The dimension of each layer is defined separately. For example the number of nodes in layer two would be defined as *two_dim***.

| hyperparameter | type | min (or options) | max (or none) |
| --- | --- | --- | --- |
| steps | randint | 50 | $10^4$ |
| lag* | randint | 1 | 2 |
| batch | choice | {-1, 50-100} | · |
| lr | loguniform | $10^{-5}$ | $10^{-1}$ |
| num_layers | randint | 1 | 5 |
| {one-five}_dim** | randint | 1 | 10 |
| grid | randint | 1 | 100 |
| k | randint | 1 | 3 |
| $\lambda$ | loguniform | $10^{-7}$ | $10^{-3}$ |
| $\lambda_{coef}$ | loguniform | $10^{-7}$ | $10^{-3}$ |

Table 22: Hyperparameter search space for the KAN model on the KS system. NOTE: The lag parameter is set to zero for reconstruction tasks (pair_id = 2 or 4)*. The dimension of each layer is defined separately. For example the number of nodes in layer two would be defined as *two_dim***.

Here, $u_\theta(x, t)$ denotes a neural network approximation of the solution with fitting parameters $\theta$, and independent variable inputs $(x, t)$. $u(x, t)$ is the ground truth at data points $(x, t)$, and $\mathcal{N}_\gamma[u] = 0$ represents the residual, with differential operator $\mathcal{N}_\gamma$ and fitting model parameters $\gamma$. The first term, $\mathcal{L}$data, ensures agreement with observed data (e.g., initial and boundary conditions), while the second term, $\mathcal{L}$DE, enforces consistency with the known physical laws through collocation points.

PINNs were originally designed as differential equation solvers [35], and they excel at interpolating solutions within a domain where collocation points are defined. Their primary strength lies in approximating solutions to known equations. While they can, in principle, be extended to infer unknown parameters of the governing equations by treating them as learnable variables in the loss function, this joint optimization (i.e. over both the neural network parameters $\theta$ and the model parameters $\gamma$) is notoriously difficult. In complex spatio-temporal settings, this often leads to poor convergence and suboptimal solutions, as observed in our CTF. Recent extensions show promising directions for improvement [67, 14].

**Implementation.** We use the DeepXDE library [49] to implement the PINN architecture, building on the inverse modeling example provided for the Lorenz system [50]. In our implementation, we assume a parametric form of the target differential equation (e.g., Lorenz or Kuramoto–Sivashinsky) and treat all coefficients as learnable parameters.

**Hyperparameters.** Our hyperparameter search includes the learning rate, network depth and width, and the number of training, boundary, and collocation points used to evaluate the data and physics loss terms. Table 23 summarizes the hyperparameter search space.

| hyperparameter | type | range (or options) |
|---|---|---|
| Number of layers | integer | $[3, 6]$ |
| Number of neurons per layer | integer | $[10, 40]$ |
| Number of boundary points | integer | $[200, 1000]$ |
| Number of domain points (for PDE) | integer | $[200, 1000]$ |
| Learning Rate | loguniform | $[10^{-5}, 10^{-2}]$ |

Table 23: Hyperparameter search space for PINNs.

## 4.12 Neural-ODE

Nerual-ODEs are a type of neural network that uses an ODE solver to model the hidden state of a neural network.[12]. This is very similar to ODE-LSTMs, another model evaluated in this work, except it makes use of a vanilla MLP instead of LSTM.

We search over the following hyperparameters: hidden_state_size (dimension of the latent space), seq_length (input sequence length), batch size, and lr (learning rate).

| hyperparameter | type | min (or options) | max (or none) |
|---|---|---|---|
| hidden_state_size | randint | 8 | 1024 |
| seq_length | randint | 5 | 74 |
| batch_size | randint | 5 | 120 |
| lr | log_uniform | $10^{-5}$ | $10^{-2}$ |

Table 24: Hyperparameter search space for Neural-ODE models. We train for 100 epochs.

## 4.13 LLMTime

LLMTime [20] is a time-series foundation model that uses pre-trained LLMs to perform zero-shot forecasting of time-series data. Their approach is to modify the tokenization of each model so that time-series forecasting is casted as a next-token prediction in text problem. For our evaluation, we used the `llama-7b` as LLMTime's base LLM and used the default temperature of 1.0, an alpha of 0.99, and a beta of 0.3. We also used LLMTime's default Llama tokenizer. LLMTime is only able to forecast univariate time-series, so we auto-regressively forecast each dimension with a context of 200 tokens and a prediction length of 100 tokens at a time. Once each dimension has been forecasted, they are concatenated and evaluated on the test set. For reconstruction tasks, we take the first 10 time-steps of the training data and forecast each dimension until we have a vector containing the same number of timesteps as in the testing dataset and then concatenate and calculate our metrics as before.

## 4.14 Chronos

Chronos [3] is a pre-trained probabilistic time-series foundation model from Amazon. The model is informed by the success of transformers and LLMs, and as such tokenizes time series values using scaling and quantization and trains using the cross-entropy loss function. The model is only capable of doing univariate time-series forecasting. For our evaluation, we use the pre-trained `chronos-t5-base` model and do a one-shot forecast of each dimension of each dataset independently and concatenate them when calculating our metrics. For reconstruction tasks, we take the first 10 time-steps of the training data and forecast each dimension until we have a vector containing the same number of timesteps as in the testing dataset and then concatenate and calculate our metrics as before. Chronos has a much smaller context length than LLMTime due to requiring more VRAM for inference.

## 4.15 Moirai

Moirai_MoE [43] is a time-series forecasting foundation model from Salesforce AI Research. The model uses a sparse mixture-of-experts transformer architecture and is able to do one-shot multivariate time-series forecasting on arbitrary time-series datasets. For our evaluation, we used the pre-trained

`base` model and predicted 10 time-steps at a time with a context length of 20. For reconstruction tasks, we take the first 10 time-steps of the training data and forecast until we have a matrix containing the same number of timesteps as the testing dataset. Moirai_MoE has a much smaller context length than LLMTime due to requiring more VRAM for inference.

### 4.16 Sundial

Sundial [44] is a family of native, flexible and scalable time-series foundation models from Tsinghua University, tailored specifically for time series analysis. It is pre-trained on TimeBench (about one trillion time points), adopting a flow-matching approach rather than fixed parametric densities. Sundial directly models the distribution of next-patch values in continuous time-series without discrete tokenisation; it is built on a decoder-only Transformer architecture. For our evaluation, we used the pre-trained `sundial-base-128m` model; the model can handle multivariate time-series forecasting directly. For the KS evaluation, due to RAM limitations, we have split the "spatial" dimension into batches, forecasting each batch independently and concatenating the results. For reconstruction tasks, we take some of the first time-steps of the training data (around 10%) and forecast until we have a matrix containing the same number of timesteps as the testing dataset.

### 4.17 Panda

Panda [36] is a foundation model for nonlinear dynamical systems based on Patched Attention for Nonlinear DynAmics. Panda is motivated by dynamical systems theory and adopts an encoder-only architecture with a fixed prediction horizon. It is pre-trained purely on a synthetic dataset of $2 \times 10^4$ chaotic dynamical systems, discovered using a structured algorithm for dynamic systems discovery introduced in the same work. For our evaluation, we used the pretrained model weights provided on the official code repository associated with [36]. The main free parameter in the forecasts with Panda is the context length. In the Lorenz evaluation we allow this to be the full dataset that we provide, but due to RAM limitations for the KS dataset we have to limit the context to 512 observations.

### 4.18 TabPFN-TS

TabPFN for Time Series (TabPFN-TS) [26] is based on the tabular foundation model TabPFNv2 [25], adapted to the task of time series forecasting. We use the pretrained model weights, leaving the only remaining parameter as the amount of data for each specific system that the model is exposed to before performing zero-short forecasting. In the case of the Lorenz system, this is the entirety of the available training data for the task. However, for the KS system, we restrict to at most 500 time steps to be used for context. This restriction was introduced as a result of limited available memory, and is similar to the restriction placed on Panda.