# OpenReview forum: "Common Task Framework For a Critical Evaluation of Scientific Machine Learning Algorithms"
_NeurIPS.cc/2025/Datasets_and_Benchmarks_Track — NeurIPS 2025 Datasets and Benchmarks Track poster_

### Official Review · Reviewer_ToG7 · 2025-06-28

**Rating:** 5
**Confidence:** 4

**Summary:**

This paper proposes a common task framework (CTF) for scientific machine learning problems. The authors provide a benchmarking platform with standardized datasets for two problems -- a one-dimensional spatio-temporal PDE and the Lorenz dynamical system. Moreover, they design diverse evaluation tasks and multiple metrics for a comprehensive evaluation of algorithms over baselines. The authors further report the performance of different models with clear metric numbers.

**Additional Feedback:**

This paper contributes to the filed of AI4Science. However, the scope of this paper is a bit limited for now -- it only considers one specific PDE and one dynamical system problem. It would be great if the authors could expand the datasets to handle more problems, or maybe explain the major challenges of not having more science problems in the dataset.

**Dataset Code Accessibility:**

Yes

**Dataset Code Comments:**

The datasets and code are provided on github and kaggle. They are well documented and maintained.

**Ethical Comments:**

This paper falls into the AI4Science category, and I do not see any ethical concerns.

**Ethical Considerations:**

No, there are no or only very minor ethics concerns

**Final Justification:**

1. The connections between dynamical systems and modern ML systems deserve more attention, and this work makes a meaningful step towards bridging the gap between these two areas.

2. The authors added a new dataset to further expand the scope of this paper.

Therefore I would like to increase my score to accept to support them

**Limitations Weaknesses:**

1. The scope is limited -- this paper only considers two datasets. One is from a special PDE problem and the other one is from a dynamical system problem. It would be great if the authors could enrich the datasets or find other ways to generalize the datasets to more problems, otherwise it is rather a simple wrapper of datasets from two problems.

2. The performance comparison in tables lack some details.

**Strengths Contributions:**

1. This paper falls into the AI4Science category. It provides a novel framework for better evaluation of scientific machine learning methods. The datasets and code are well documented and maintained.

2. Evaluations of scientific machine learning algorithms are non-trivial. For example the dynamical system problems are typically accompanied by chaotic behaviors. This paper provides rigorous metrics to benchmark ML algorithms to tackle these challenges.

---

> ### Author Rebuttal · Authors · 2025-07-31
>
> # General Response
> Thank you for your careful review of our manuscript and for your valuable comments and questions. We appreciate your suggestions regarding expanding the benchmark to include more datasets and models. We would like to take this opportunity to clarify the primary intent of our work. We hope that our response adequately answers your questions and address your concerns, and thereby encourages you to consider raising your score for acceptance.
>
> The central goal of the proposed Common Task Framework (CTF) is to establish a standardized infrastructure (including a unified interface, evaluation protocols, and metric suite) for benchmarking scientific machine learning methods on problems that are often governed by PDEs and ODEs. This effort is not meant to present an exhaustive exploration of models or datasets at this initial stage. Rather, it is intended as a scalable foundation for the broader community to build upon, akin to how benchmark platforms like CASP catalyzed community-driven progress through shared, evolving tasks.
>
> The design of the CTF anticipates growth through community contribution, particularly by integrating it with platforms such as Kaggle, where both datasets and models can be crowd-sourced and rigorously compared. This approach stands in contrast to self-reported benchmarks, which are often unreliable and do not provide fair or reproducible assessments.
>
> Importantly, the key finding from our initial experiments is that state-of-the-art scientific ML methods often fail to perform well; even on problems that are traditionally considered “simple.” Our benchmark results (see Figure 3 and Table 1 in the manuscript) show that many leading models underperform relative to trivial baselines. This result itself highlights the urgent need for a rigorous, community-wide benchmark: if we cannot yet reliably forecast systems like Lorenz or Kuramoto–Sivashinsky (KS), we should not expect robust performance in more complex real-world applications such as climate or materials modeling.
>
> This mirrors the historical trajectory of successful CTFs like CASP, which began with relatively simple protein folding tasks and progressively moved to more challenging problems as methods matured. Similarly, our approach begins with foundational systems and will evolve with community input.
>
> In response to the reviewer’s suggestion, we have already taken a step toward expansion by incorporating a new real-world, high-resolution spatiotemporal dataset: daily sea-surface temperature (SST) observations from NASA. This dataset, extracted from an undisclosed time and location, reflects the kind of complex, structured scientific data we intend to support. While full model results are not yet available due to the extensive data preparation and hyperparameter tuning required, this dataset is fully prepared and will be launched alongside the upcoming Kaggle competition.
>
> We also note that while our initial benchmark includes only two dynamical systems (Lorenz and KS), these are far from trivial. Despite their low dimensionality, they exhibit rich, chaotic behavior that continues to challenge modern ML models. In fact, their deceptive simplicity makes them ideal candidates for highlighting the weaknesses and limitations of current methods. This justifies our focus on these systems at the current stage.
>
>
> Finally, we emphasize that the benchmark is deliberately limited at launch to avoid overwhelming contributors with too many datasets and to ensure clarity of evaluation. The current selections represent diverse and representative dynamics encountered in scientific ML and offer a meaningful starting point for iterative community-driven expansion.
>
>   # Responses to Individual Questions/Observations
>
>
> **1.  The scope is limited -- this paper only considers two datasets. One is from a special PDE problem and the other one is from a dynamical system problem. It would be great if the authors could enrich the datasets or find other ways to generalize the datasets to more problems, otherwise it is rather a simple wrapper of datasets from two problems.**
>
>
> Thank you for highlighting this limitation. We partially address your concern in the general response above. This being said, we agree that expanding the benchmark is important for broader scientific relevance. In response, we have added a real-world 2D spatiotemporal dataset: daily sea-surface temperature (SST) data from NASA, extracted from satellite observations. This dataset introduces: 1) Multiscale temporal structure, 2) High-dimensional spatial variability, 3) Noisy and incomplete experimental measurements.
>
> This dataset will be included in our upcoming Kaggle launch, allowing participants to test models on data that more closely reflects real scientific complexity. As mentioned in the general response, our initial dataset selection (Lorenz and KS) is intentional. Despite being considered "simple," these systems exhibit chaotic and nonlinear dynamics that challenge current ML methods. Starting with interpretable, canonical systems also makes it easier to diagnose failure modes, compare methods rigorously, and build a shared foundation before moving to more complex, domain-specific settings.
>
>
> **2.  The performance comparison in tables lack some details.**
>
>
> We have added a summary table to the main manuscript, listing the training and test windows for each forecasting task, and mapping these explicitly to their corresponding score definitions (E1–E12). Included details on dataset size, shape, and file structure for clarity and reproducibility. Furthermore, we’ve referenced the supplementary material more clearly, where further implementation details and hyperparameter tuning strategies are outlined. These improvements aim to make the benchmark easier to understand and extend, both for replication and for community contributions.
>
>
> **3.  This paper contributes to the field of AI4Science. However, the scope of this paper is a bit limited for now -- it only considers one specific PDE and one dynamical system problem. It would be great if the authors could expand the datasets to handle more problems, or maybe explain the major challenges of not having more science problems in the dataset.**
>
>
> Thank you for this observation. As discussed in the general response above, we have now added a real-world, experimental dataset, and we plan to actively expand the benchmark in future iterations. However, scaling up comes with nontrivial challenges, including:
>
>
> -   Hyperparameter tuning costs: Many models require substantial tuning to provide fair competitive results, particularly on high-dimensional data.
>
>
> -   Compute resources: Running fair and reproducible comparisons at scale is computationally intensive and requires consistent access to high-performance GPUs.
>
>
> -   Metric standardization: Scientific problems vary widely in characteristics; developing generalizable metrics that work across stochastic, multiscale, and coupled systems remains an open challenge.
>
>
> Nonetheless, with the infrastructure now in place, we are confident the CTF can evolve into a robust community benchmark for AI4Science—serving both as a diagnostic tool and as a platform for advancing new model architectures and evaluation protocols.

---

> > ### Author Response · Authors · 2025-08-06
> >
> > Dear reviewer,
> >
> > We would like to thank you again for your valuable initial feedback and engagement with our work.
> >  We believe our rebuttal above has addressed the limitations and weaknesses that you mentioned, and we would greatly appreciate if you could respond and let us know your thoughts and if there are any further concerns that we could address. If you feel the concerns have been addressed, we would appreciate it if you would consider increasing your score. Thank you for your time and engagement with our work.

---

> > > ### Author Response · Authors · 2025-08-08
> > >
> > > Dear reviewer,
> > >
> > > As we enter the final hours of the discussion period, we would appreciate hearing your thoughts on our rebuttal.  If you feel the concerns have been addressed, we would appreciate it if you would consider increasing your score.

---

> > ### Comment · Reviewer_ToG7 · 2025-08-08
> > **Response to the rebuttal**
> >
> > Dear authors,
> >
> > Thank you for your efforts in the rebuttal. My questions are in general addressed. A few minor comments.
> >
> > 1. "Despite their low dimensionality, they exhibit rich, chaotic behavior that continues to challenge modern ML models." I agree ML systems can have chaotic behaviors and there are actually recent works on this. It would be good if the authors could have a brief discussions on them ([1][2][3][4]) to support their claim on the relationship between dynamical system theory and ML.
> >
> > 2. Thanks for adding the SST dataset. I understand the difficulties of further expanding the scope of this paper.
> >
> > I would like to increase score to encourage the authors to continue to work on this direction.
> >
> > References:
> > [1] The boundary of neural network trainability is fractal
> >
> > [2] Stochasticity of deterministic gradient descent: Large learning rate for multiscale objective function
> >
> > [3] From stability to chaos: Analyzing gradient descent dynamics in quadratic regression
> >
> > [4] Universal Sharpness Dynamics in Neural Network Training: Fixed Point Analysis, Edge of Stability, and Route to Chaos

---

> > > ### Author Response · Authors · 2025-08-08
> > >
> > > Thank you for your response! We will include those references. We appreciate your engagement with our work and thank you for raising your score.

---

### Official Review · Reviewer_nupn · 2025-07-01

**Rating:** 4
**Confidence:** 3

**Summary:**

This paper proposes a Common Task Framework (CTF) for evaluating scientific machine learning models. Inspired by benchmarks like ImageNet and GLUE, the authors present a benchmark suite with 12 metrics, a composite score, and two canonical datasets (Lorenz ODE and Kuramoto-Sivashinsky PDE). The goal is to enable standardized, rigorous comparisons of modeling methods across forecasting, reconstruction, and generalization tasks under challenging conditions such as noise and limited data.

**Dataset Code Accessibility:**

Yes

**Ethical Considerations:**

No, there are no or only very minor ethics concerns

**Final Justification:**

My concerns have been addressed.

**Limitations Weaknesses:**

1. The datasets are small in size (by design), limiting generalization claims.  The current benchmark does not represent the complexity or diversity of scientific systems encountered in practice (e.g., multi-scale PDEs, high-dimensional data, experimental observations).
2. The proposed 12-metric composite score spans only two physical systems. Scores may not generalize to different scientific regimes (e.g., stochastic, multi-scale, or coupled systems)
3.  No Real-World or Experimental Data.
4. Many models are reused from previous implementations, and the paper does not:  (1) Explain in detail how hyperparameters were selected. (2) Discuss model adaptation difficulties (e.g., why DeepONet struggles on autoregressive tasks). (3) Show robustness via repeated runs or error bars

**Strengths Contributions:**

1.  The problem is timely and relevant: standardized evaluation is essential for scientific reproducibility in SciML.

---

> ### Author Rebuttal · Authors · 2025-07-31
>
> # General Response
> Thank you for your careful review of our manuscript and for your valuable comments and questions. We appreciate your suggestions regarding expanding the benchmark to include more datasets and models. We would like to take this opportunity to clarify the primary intent of our work. We hope that our response adequately answers your questions and address your concerns, and thereby encourages you to consider raising your score for acceptance.
>
>
>
> The central goal of the proposed Common Task Framework (CTF) is to establish a standardized infrastructure (including a unified interface, evaluation protocols, and metric suite) for benchmarking scientific machine learning methods on problems that are often governed by PDEs and ODEs. This effort is not meant to present an exhaustive exploration of models or datasets at this initial stage. Rather, it is intended as a scalable foundation for the broader community to build upon, akin to how benchmark platforms like CASP catalyzed community-driven progress through shared, evolving tasks.
>
>
>
> The design of the CTF anticipates growth through community contribution, particularly by integrating it with platforms such as Kaggle, where both datasets and models can be crowd-sourced and rigorously compared. This approach stands in contrast to self-reported benchmarks, which are often unreliable and do not provide fair or reproducible assessments.
>
>
>
> Importantly, the key finding from our initial experiments is that state-of-the-art scientific ML methods often fail to perform well; even on problems that are traditionally considered “simple.” Our benchmark results (see Figure 3 and Table 1 in the manuscript) show that many leading models underperform relative to trivial baselines. This result itself highlights the urgent need for a rigorous, community-wide benchmark: if we cannot yet reliably forecast systems like Lorenz or Kuramoto–Sivashinsky (KS), we should not expect robust performance in more complex real-world applications such as climate or materials modeling.
>
>
>
>
> This mirrors the historical trajectory of successful CTFs like CASP, which began with relatively simple protein folding tasks and progressively moved to more challenging problems as methods matured. Similarly, our approach begins with foundational systems and will evolve with community input.
>
>
>
> In response to the reviewer’s suggestion, we have already taken a step toward expansion by incorporating a new real-world, high-resolution spatiotemporal dataset: daily sea-surface temperature (SST) observations from NASA. This dataset, extracted from an undisclosed time and location, reflects the kind of complex, structured scientific data we intend to support. While full model results are not yet available due to the extensive data preparation and hyperparameter tuning required, this dataset is fully prepared and will be launched alongside the upcoming Kaggle competition.
>
>
>
> We also note that while our initial benchmark includes only two dynamical systems (Lorenz and KS), these are far from trivial. Despite their low dimensionality, they exhibit rich, chaotic behavior that continues to challenge modern ML models. In fact, their deceptive simplicity makes them ideal candidates for highlighting the weaknesses and limitations of current methods. This justifies our focus on these systems at the current stage.
>
>
>
> Finally, we emphasize that the benchmark is deliberately limited at launch to avoid overwhelming contributors with too many datasets and to ensure clarity of evaluation. The current selections represent diverse and representative dynamics encountered in scientific ML and offer a meaningful starting point for iterative community-driven expansion.
>
>  # Responses to Individual Questions/Observations
>
>
> **1.  The datasets are small in size (by design), limiting generalization claims. The current benchmark does not represent the complexity or diversity of scientific systems encountered in practice (e.g., multi-scale PDEs, high-dimensional data, experimental observations).**
>
>
>
>
> While the main purpose of the framework is discussed in the general response above, we agree that scientific systems often involve multi-scale, high-dimensional, and stochastic behaviors. Our initial datasets (Lorenz, KS) were chosen intentionally to create interpretable and controlled testbeds where fundamental modeling challenges (such as instability, chaotic dynamics, and memory retention) can be precisely evaluated.
>
> To address this limitation, we have now added a new real-world dataset of daily sea-surface temperature (SST) observations, derived from NASA satellite measurements. This dataset is high-dimensional and spatiotemporal, contains multi-scale temporal patterns, and is grounded in experimental observations.
>
>
>
> **2.  The proposed 12-metric composite score spans only two physical systems. Scores may not generalize to different scientific regimes (e.g., stochastic, multi-scale, or coupled systems)**
>
>
>
>
> You are correct that the scores are specific to the evaluated systems. However, the metrics themselves are designed to be dataset-agnostic: they reflect evaluation priorities that scientists care about across many modeling regimes, such as:
>
> -   Reconstruction quality
>
> -   Long-range prediction stability
>
> -   Robustness to limited data
>
> -   Parameter extrapolation/generalization
>
>
>
>
> Indeed, we do not expect methods to generalize across datasets, and this is a core motivation for the CTF: to provide a common infrastructure where new models can be tested across different scientific domains without redefining the task or metrics every time.
>
>
>
> **3.  No Real-World or Experimental Data.**
>
>
>
>
> As noted above, we have added real-world, high-dimensional and multiscale NASA sea-surface temperature (SST) data that was collected via satellite instrumentation. This dataset is prepared and will be released as part of the Kaggle launch. We are going to include more real-world datasets as the Common Task Framework evolves.
>
>
>
> **4.  Many models are reused from previous implementations, and the paper does not: (1) Explain in detail how hyperparameters were selected. (2) Discuss model adaptation difficulties (e.g., why DeepONet struggles on autoregressive tasks). (3) Show robustness via repeated runs or error bars**
>
>
>
>
> We appreciate this important point and have made the following updates:
>
>
>
> -   Hyperparameter tuning: we conducted grid searches or guided tuning for all methods. Please refer to the supplementary material for more details, and we have now added explicit references to these details in the main manuscript.
>
>
>
>
> -   Model adaptation difficulties: Some models (like DeepONet) require significant adaptation for autoregressive forecasting tasks. We have method-specific comments on such adaptation challenges in the supplementary material, including issues of architectural changes, memory bottlenecks, or training instability. Given space constraints, these details were placed in the supplement but are critical and now clearly referenced.
>
>
>
>
> -   Robustness and error bars: Thank you for this suggestion. We have now included standard deviations across multiple training runs in the updated result tables. This provides a clearer picture of the robustness and variability of model performance.
>
>
>
> “Models are reused from previous implementations”: This is a very crucial point to note: the main contribution of our work is the evaluation and benchmark dataset and framework, which is why we made an effort to stay faithful to models from prior implementations, to try and evaluate models as fairly as possible. In addition, we have
>
> 1.  Selection of hyperparameters: we provided full details on the selection of hyperparameters for all models in the appendix, we did not mention this in the main part of our submission, but added it now. In fact, for each model hyperparameter we provide a sensible range of hyperparameters which we then explore using Ray Tune on an analogous challenge with slightly modified data (for fairness). The optimal set of hyperparameters are then used in the final evaluation of each model. We will clarify this in the main part of the camera ready version of the manuscript.
>
> 2.  Model adaption challenges: Understanding why certain models perform better on specific tasks than others is an important question for future work, however the main focus of this contribution is to provide the CTF challenge and to objectively evaluate models on this challenge without judging or interpreting these results. We note, however, that the challenge does highlight how certain models may be more suitable for some tasks (e.g. noisy or incomplete data) while performing worse in other settings (e.g. long-term predictions). This nuanced performance is highlighted in our radar plots, thus effectively allowing domain specialists to use our CTF to inform their model choice based on the task at hand.
>
> 3.  Repeated runs and error bars: We agree that repeated runs and the computation of error bars (standard deviation) is incredibly important. We have commenced additional computational experiments to be included in the final submission, evaluating each model multiple times on the same dataset to understand their average performance and variability. Preliminary results for the top 5 models on the Lorenz Dataset are reported in the table below, and full results will be included in the camera-ready version.
>
> **Table: Mean and standard deviation in composite score for the top 5 models on the Lorenz Dataset**
>
> | Model name  | Average Score | Standard Deviation |
> |-------------|---------------|--------------------|
> | SINDy       |     63.27     |        0.00        |
> | Reservoir   |     54.59     |        2.88        |
> | PyKoopman   |     49.96     |        0.00        |
> | DeepONet    |     44.46     |       12.36        |
> | FNO         |     35.05     |       10.00        |

---

> > ### Comment · Reviewer_nupn · 2025-08-07
> >
> > Thank you so much for your detail response, my concerns have been addressed, and I will increase my score to Borderline accept.

---

### Official Review · Reviewer_zMvQ · 2025-07-02

**Rating:** 5
**Confidence:** 3

**Summary:**

The authors introduce a Common Task Framework that would provide as a standardized evaluation method in the field of scientific machine learning. The authors demonstrates examples on two dynamical systems: the Kuramoto-Sivashinsky system and the Lorenz systems. Through testing on forecasting, noisy data, limited data, and parametric generalization, the authors give difference scores to evaluate the various aspects of the models that matches real-world challenges. The authors tested on several common models to demonstrate the strength of their framework. The authors believe the ctf4science is a first step to establish a clear and robust evaluation framework for scientific machine learning models.

**Additional Feedback:**

* As the authors mentioned, the dataset size is small. Though the authors state that this is an intentional choice for the initialization of the framework, it would create risk since models can be trained to narrowly fit only on such datasets, especially for canonical dynamical systems that are well-understood and have widely available numerical solvers.

**Dataset Code Accessibility:**

Yes

**Dataset Code Comments:**

Both the datasets and the code are readily accessible. While the code documentation is clear, the dataset documentation can be improved by providing more details, such as its shape, size, and file structure.

**Ethical Comments:**

Datasets based on dynamical systems, which have no or only very minor ethics concerns.

**Ethical Considerations:**

No, there are no or only very minor ethics concerns

**Final Justification:**

This paper would be helpful if serve as an option of evaluation framework for scientific machine learning models.

* The authors addressed most of my concerns. They provided a better grouped score that offers improved explanatory capabilities, gave a reasonable explanation for their choice of models, and added results from more modern models, which demonstrated the framework's usefulness. The authors also included more detailed documentation.
* I still encourage the authors to add more diverse datasets to make it a more comprehensive framework.

Since the authors nicely addressed my concerns, I raised my score by 1.

**Limitations Weaknesses:**

* The author present 12 distinct scores, but when comparing the model scores, the authors often use the composite score. This can be misleading, since the models can perform very differently on different tasks, and a simple average hides these important details. On the other hand, the 12 different scores can be confusing when trying to compare the general performance of the models. Would it better to group and summarize some of the scores, which would provide a more robust evaluation than barely the average score, and a clearer overview beyond the very detailed radar plot?
* Also, the authors did not include the performance of some modern scientific machine learning models, for example multiple physics pretraining (MPP), In-Context Operator Networks (ICON), and DPOT. This raises the question of whether the benchmark is challenging enough for the latest methods and whether the framework is actually useful. It is possible that the modern models would achieve similarly high scores, making it hard to compare in this framework.

**Strengths Contributions:**

* The authors introduce a concrete framework to standardize the evaluation of scientific machine learning models and provide testing examples to justify their assertion. The authors provide a set of tasks and metrics, and test using canonical systems, Lorenz and Kuramoto-Sivashinsky. Such a framework can be useful to address the reproducibility crisis and enable direct comparisons in the AI for Science field.
* The paper is well-written and organized. The paper clearly lists its limitations and the vision for the future.
* The authors provide detailed code reference and clean datasets.

---

> ### Author Rebuttal · Authors · 2025-07-31
>
> # General Response
> Thank you for your thoughtful review. We hope our response addresses your concerns and encourages you to consider a higher score.
>
> The goal of our Common Task Framework (CTF) is to establish standardized infrastructure—including a unified interface, evaluation protocols, and task-specific metrics—for benchmarking scientific ML methods on problems governed by ODEs and PDEs. This first release is not exhaustive, but a scalable foundation for reproducible, community-driven evaluation, and is modeled on CASP’s early benchmarks that catalyzed progress in protein folding.
>
> Our CTF is designed for growth via crowd-sourced datasets and models, integrated through platforms like Kaggle and scored on **hidden test sets**. This contrasts with self-reported benchmarks, which often lack fairness and reproducibility.
>
> Our initial results reveal that state-of-the-art models perform worse than trivial baselines on even canonical problems like Lorenz and Kuramoto–Sivashinsky (KS) (see Fig. 3 and Table 1). This underscores the need for a rigorous community benchmark: if we cannot yet handle these, complex real-world systems like climate or materials modeling remain out of reach.
>
> In response to your suggestion, we have added a real-world dataset: daily sea-surface temperature (SST) observations from NASA. This high-resolution, structured dataset is preprocessed and will be launched with our upcoming Kaggle competition.
>
> While our initial benchmark includes only Lorenz and KS, these are not trivial. Their chaotic dynamics challenge ML methods and reveal failure modes in a controlled setting. Their simplicity makes them ideal for exposing core weaknesses.
>
> The scope is intentionally narrow at launch to ensure clarity and accessibility. Lorenz and KS span low- and high-dimensional chaos, providing a representative starting point for community expansion.
>
>   # Responses to Individual Questions/Observations
>
> **1.  The author present 12 distinct scores, but when comparing the model scores, [...] (see above for unabbreviated comment) Would it better to group and summarize some of the scores, which would provide a more robust evaluation than barely the average score, and a clearer overview beyond the very detailed radar plot?**
>
> Thank you for this suggestion. We agree that both aggregated and fine-grained views are necessary. In response, we have:
> -   Added grouped score summaries in the manuscript, clustering the 12 metrics into three categories: (1) reconstruction and forecasting (E1-E6), (2) low-data regime (E7-E10), and (3) parametric generalization (E11&E12).
> -   Provided clearer descriptions for these categories to help interpret model performance more robustly.
> -   In addition, we are building an interactive leaderboard (linked to the Kaggle competition) on our website, allowing users to select custom score groupings and explore models’ strengths and weaknesses depending on the use case. This flexibility is central to the long-term vision of the CTF.
>
> **2.  Also, the authors did not include the performance of some modern scientific machine learning models, for example multiple physics pretraining (MPP), In-Context Operator Networks (ICON), and DPOT. This raises the question whether the benchmark is challenging enough for the latest methods, and whether the framework is actually useful. It is possible that the modern models would achieve similarly high scores, making it hard to compare in this framework.**
>
> Thank you for this insightful comment. Modern SciML models, such as foundation models, are indeed very interesting and the aim of the CTF is to provide a suitable pipeline to evaluate them rigorously. To address the request for more modern models, we have now evaluated several foundation models that can perform zero-shot predictions without requiring task-specific fine-tuning on the CTF. Examples of such models include Chronos (Ansari et al., arXiv:2403.07815 (2024)), Moirai (Liu et al., arXiv:2410.10469 (2024)), TabPFN (Hollmann et al., arXiv:2207.01848 (2022)), Panda (Lai et al., arXiv:2505.13755 (2025)), Sundial (Liu et al., arXiv:2502.00816 (2025)) and LLMTime (Gruver et al., arXiv:2310.07820 (2023)), which according to the original publications perform very well in time-series prediction without fine-tuning. We will include detailed evaluation results in the final version of this paper, the preliminary results are shown below:
>
> **Table 1: Zero-shot foundation model evaluation on Lorenz Dataset**
>
> | Model name | Avg. Score |   E1   |   E2   |   E3   |   E4   |   E5   |   E6   |   E7   |   E8   |   E9   |  E10   |  E11   |  E12   |
> |------------|------------|--------|--------|--------|--------|--------|--------|--------|--------|--------|--------|--------|--------|
> | Panda      |   -45.14   | -49.61 | -49.60 | -100.0 | -47.20 | -100.0 | -26.00 | -51.48 | -29.07 |  26.42 | -43.47 | -35.88 | -35.85 |
> | Sundial    |    18.18   |  60.15 | -27.46 |  46.66 | -40.27 |  55.91 | -39.73 |  28.47 |  45.60 |  29.99 | -54.27 |  36.43 |  76.62 |
> | Moirai     |   -12.07   |  49.96 | -88.53 |  29.74 | -84.33 |  25.61 | -84.67 |  55.25 | -87.20 |  52.28 | -90.73 |  50.06 |  27.75 |
> | Chronos    |    -7.27   |  34.80 | -84.67 |  52.85 | -86.53 |  53.40 | -88.00 |  44.18 | -88.47 |  54.01 | -85.13 |  49.24 |  57.04 |
> | TabPFN     |    -2.26   |  53.01 | -87.93 |  84.18 | -88.07 |  78.94 | -85.67 |  46.71 | -86.20 |  48.83 | -88.00 |  45.57 |  51.48 |
> | LLMTime    |   -36.89   |   4.59 | -91.40 |   0.59 | -100.0 |   0.44 | -94.47 |   4.34 | -93.73 |   4.10 | -94.47 |   8.38 |   8.99 |
>
> **Table 2: Zero-shot foundation model evaluation on Kuramoto–Sivashinsky Dataset**
>
> | Model name | Avg. Score  |   E1   |   E2   |   E3   |   E4   |   E5   |   E6   |   E7   |   E8   |   E9   |  E10   |  E11   |  E12   |
> |------------|-------------|--------|--------|--------|--------|--------|--------|--------|--------|--------|--------|--------|--------|
> | Panda      |   -75.43    |  46.83 | -100.0 | -100.0 | -100.0 | -100.0 | -100.0 | -100.0 | -100.0 | -100.0 | -100.0 | -42.50 |  -9.44 |
> | Sundial    |    -7.03    |   0.86 |   0.02 |   4.60 |   0.00 |   5.75 |   0.01 |   1.06 |   0.03 |  -1.53 |   0.03 | -49.71 | -45.43 |
> | Moirai     |   -93.79    | -100.0 | -100.0 | -25.53 | -100.0 | -100.0 | -100.0 | -100.0 | -100.0 | -100.0 | -100.0 | -100.0 | -100.0 |
> | Chronos    |   -23.03    |  37.89 |  26.91 | -100.0 |  -6.24 | -100.0 |   3.44 |  -4.11 |   0.21 | -23.40 | -100.0 |  -7.02 |  -4.08 |
>
>
> We note that for two models (TabPFN and LLMTime), the memory and computational requirements of evaluation on the KS dataset were prohibitive with the limited time available. For those models, we only show preliminary results for the Lorenz system.
>
> We have also considered the proposed models mentioned by the reviewer:
>
>  - ICON is capable of learning new operators without fine-tuning, which fits the zero-shot paradigm well. However, to the best of our knowledge, the pre-trained weights and corresponding datasets used in the paper are not publicly available. As a result, replicating the ICON training pipeline (which involves generating a large number of operator snapshots and training a transformer from scratch) is currently unfeasible for time and resource reasons.
>
>  - DPOT is a pre-training strategy for a transformer modeling multiple PDEs. Nevertheless, it requires fine-tuning on downstream tasks, which diverges from our current emphasis on zero-shot evaluation. That said, we believe DPOT and similar strategies could be excellent candidates for a future extension of the benchmark.
>
>  - Multiple physics pretraining (MPP) is another example of a pretraining approach for learning PDEs. Much like DPOT, MPP can be evaluated in a zero-shot manner, but the authors’ intended and explored use of the method involves fine-tuning. As a result, we were unable to fairly evaluate MPP for time and resource reasons.
>
> We fully agree that extending the benchmark to cover a broader range of foundation models is important and we are currently exploring this direction, both with and without fine-tuning.
>
> As discussed in the general response above, the purpose of the CTF is to establish the infrastructure and evaluate the fundamental capability of models to generalize across tasks. The fact that many advanced (e.g. foundational time series and SciML) models struggle even in this zero-shot regime highlights the utility and challenge of the benchmark.
>
> **3.  the dataset documentation can be improved by providing more details, such as its shape, size, and file structure.**
>
> We appreciate this suggestion. To improve clarity, we have added a summary table in the main manuscript listing the training and test ranges for each forecasting task and explicitly matching score definitions (E1, E2, etc.) to their respective data windows. We’ve also included more details on the size, shape and file structure.
>
> **4.  As the authors mentioned, the dataset size is small. Though the authors state that this is an intentional choice for the initialization of the framework, it would create risk since models can be trained to narrowly fit only on such datasets, especially for canonical dynamical systems that are well-understood and have widely available numerical solvers.**
>
> We appreciate this concern. As we noted in the general response, the decision to start with compact, well-understood systems like Lorenz and KS was intentional. Despite being “simple”, these systems are nonlinear and chaotic, and many modern methods still fail to perform reliably on them. Their interpretability and availability of analytical baselines make them ideal testbeds for early benchmark development. That said, in response to reviewer comments, we are including a larger, real-world dataset. This will help test methods at larger scales while maintaining continuity with the original benchmark structure.

---

> > ### Comment · Reviewer_zMvQ · 2025-08-05
> >
> > Dear authors, thank you for your response. My concerns have been addressed, and I will increase my score by 1.

---

> > > ### Author Response · Authors · 2025-08-06
> > >
> > > Thank you for your valuable feedback and engagement with our work.

---

### Official Review · Reviewer_qUAq · 2025-07-03

**Rating:** 5
**Confidence:** 4

**Summary:**

The authors present ctf4science, a common task framework that enables multi-dimensional evaluation of scientific ML models. Currently, the authors have prepared this for two nonlinear systems: Kuramoto-Sivashinsly and Lorenz. Different ML models are evaluated on a set of 12 different tasks. While currently limited to two systems, the paper has a much broader impact and serves as a foundation for future CTFs which would allow the field to develop better ML models that can be compared in a consistent manner instead of self-reported cherry picked scores.

**Dataset Code Accessibility:**

Yes

**Dataset Code Comments:**

The code and data provided are clear, accessible and easy to use. I cloned the repo and ran some benchmarks myself. The provided instructions are clear and worked without much manual intervention. The notes in the repo also provide more details about the error metrics and have a few more systems other than the two described in the paper. The provided datasets are also well-documented on Kaggle and conducive to future competitions.

**Ethical Considerations:**

No, there are no or only very minor ethics concerns

**Final Justification:**

After reading the authors’ rebuttal, I am satisfied that my main concerns have been addressed. The inclusion of SST dataset makes this a much stronger benchmark for PIML. I appreciate the inclusion of related benchmarks, concise table for errors and the amortized computational score in the appendix.

The explanation for E1’s strictly positive range is reasonable, though I still encourage the authors to test whether future datasets might warrant rescaling the composite score.

These edits are not yet reflected in the uploaded manuscript and supplementary file, and assuming they will be incorporated, I keep my score at 5.

**Limitations Weaknesses:**

While there are no glaring (unaddressed) limitations in this papers, I have a few questions/comments:

1. The introduction section could have more details about existing ML-PDE benchmarks. The Well and CoDBench are mentioned in the repo but not in the paper. Some more examples could also be included like PDEBench and PDEArena. The authors should also highlight how the proposed CTF compares to these existing benchmark suites.
2. The numbering scheme for training, prediction indices can be a bit hard to parse. This is somewhat mitigated by the input, output, scores summary at the end of each section. Maybe a table might make it even more clearer which train, prediction and error indices belong to which task.
3. Related to that, there seems to be a typo for burn-in matrix indices in section 2.1.4 on page 5. Shouldn't it be X_{9train} and X_{10train}?
4. While majority of errors are in a range of [-100, 100], E1 seems to be in a range of [0, 100]. Should this be accounted for by rescaling it when calculating the composite score?
5. Maybe this might be unfeasible to add to the metrics at this stage, but can the authors add some comments on the importance of also calculating computational resources used for training/inference and by numerical methods. One ammortized metric is provided in the already cited McGreivy and Hakim paper.

**Strengths Contributions:**

The authors provide a good framework for comparing various ML models. The paper is well-written and many different aspects of issues plaguing the field of scientific ML (especially ML PDE solvers) are thoughtfully considered and addressed.

The need for a ctf in the field is contexutalized by pointing to the effectiveness of ctfs in other fields such as CV and NLP.

The chosen systems (Lorenz for ODE) and (KS for PDE) are representative of hard problems in scientific ML and the error metrics developed here can serve as templates for other systems. The split between short term direct solution error(weather) and long term statistical error (climate) serves as a good choice to show the performance of ML models on longer rollout periods.

The metrics defined in section 2 cover various scenarios that are common in scientific and engineering applications, ie standard forecasting, noise, (lack of) data availability and parameter generalization.

The metrics are validated on multiple different ML models and baselines. The observations section also covers how the metrics can show strengths and weaknesses of the models, even previously unknown ones such as  DeepONet on temporal auto-regression.

The provided code/data repo is excellent and easy to run. The systems chosen are also accessible and can be evaluated on low resource machines without gpus.

---

> ### Author Rebuttal · Authors · 2025-07-31
>
> # General Response
> Thank you for your careful review of our manuscript and for your valuable comments and questions. We appreciate your suggestions regarding expanding the benchmark to include more datasets and models. We would like to take this opportunity to clarify the primary intent of our work. We hope that our response adequately answers your questions and address your concerns, and thereby encourages you to consider raising your score for acceptance.
>
> The central goal of the proposed Common Task Framework (CTF) is to establish a standardized infrastructure—including a unified interface, evaluation protocols, and metric suite—for benchmarking scientific machine learning (ML) methods on problems that are often governed by PDEs and ODEs. This effort is not meant to present an exhaustive exploration of models or datasets at this initial stage. Rather, it is intended as a scalable foundation for the broader community to build upon, akin to how benchmark platforms like CASP catalyzed community-driven progress through shared, evolving tasks.
>
> The design of the CTF anticipates growth through community contribution, particularly by integrating it with platforms such as Kaggle, where both datasets and models can be crowdsourced and rigorously compared. Our CTF approach with models scored on hold-out datasets stands in contrast to self-reported benchmarks, which are often unreliable and do not provide fair or reproducible assessments. To drive community engagement, we are going to launch cash-prize Kaggle contests around real-world datasets, organize workshops at notable conferences (ICML, NeurIPS), and will leverage our network of peer NSF AI Institutes to drive initial adoption.
>
> Importantly, the key finding from our initial experiments is that state-of-the-art scientific ML methods often fail to perform well; even on problems that are traditionally considered “simple.” Our benchmark results (see Figure 3 and Table 1 in the manuscript) show that many leading models underperform relative to trivial baselines. This result itself highlights the urgent need for a rigorous, community-wide benchmark: if we cannot yet reliably forecast systems like Lorenz or Kuramoto–Sivashinsky (KS), we should not expect robust performance in more complex real-world applications such as climate or materials modeling.
>
> This mirrors the historical trajectory of successful CTFs like CASP, which began with relatively simple protein folding tasks and progressively moved to more challenging problems as methods matured. Similarly, our approach begins with foundational systems and will evolve with community input.
>
> In response to the reviewer’s suggestion, we have already taken a step toward expansion by incorporating a new real-world, high-resolution spatiotemporal dataset: daily sea-surface temperature (SST) observations from NASA. This dataset, extracted from an undisclosed time and location, reflects the kind of complex, structured scientific data we intend to support. While the complete list of model scores on this new dataset are not yet available due to the extensive data preparation and hyperparameter tuning effort required, the dataset is prepared and will be launched alongside our upcoming Kaggle competition.
>
> We also note that while our initial benchmark includes only two dynamical systems (Lorenz and KS), these are far from trivial. Despite their low dimensionality, they exhibit rich, chaotic behavior that continues to challenge modern ML models. In fact, their deceptive simplicity makes them ideal candidates for highlighting the weaknesses and limitations of current methods. This justifies our focus on these systems at the current stage.
>
> Finally, we emphasize that the benchmark is deliberately limited at launch to avoid overwhelming contributors with too many datasets and to ensure clarity of evaluation. The current selections represent diverse and representative dynamics encountered in scientific ML and offer a meaningful starting point for iterative community-driven expansion.
> # Responses to Individual Questions/Observations
>
> **1. The introduction section could have more details about existing ML-PDE benchmarks. The Well and CoDBench are mentioned in the repo but not in the paper. Some more examples could also be included like PDEBench and PDEArena. The authors should also highlight how the proposed CTF compares to these existing benchmark suites.**
>
> Thank you for pointing this out. We have updated the Introduction section to include references and descriptions of existing benchmarks such as The Well, CoDBench, PDEBench, and PDEArena, with corresponding citations. While some of these were mentioned in the GitHub repository, they are now properly cited and discussed in the paper. Our work proposes a Common Task Framework (CTF) designed as an extensible infrastructure for temporal and spatio-temporal forecasting tasks arising across scientific ML applications. As we elaborate in the general response above, this is intended as a community-driven infrastructure, not just a dataset/model suite. CTF defines standard interfaces, evaluation protocols, and metrics, etc. intended to evolve through community participation via Kaggle competitions and contributions.
>
> **2. The numbering scheme for training, prediction indices can be a bit hard to parse. This is somewhat mitigated by the input, output, scores summary at the end of each section. Maybe a table might make it even clearer which train, prediction and error indices belong to which task.**
>
> We appreciate this suggestion. To improve clarity, we have added a summary table in the main manuscript listing the training and test ranges for each forecasting task and explicitly matching score definitions (E1, E2, etc.) to their respective data windows. While the textual summaries at the end of each section were helpful, we agree that the new table offers a clearer and quicker reference.
>
> **3. Related to that, there seems to be a typo for burn-in matrix indices in section 2.1.4 on page 5. Shouldn't it be $X_{9train}$ and $X_{10train}$?**
>
> Thank you for catching this. You are correct—the correct indices are X_{9train} and X_{10train}. This has been corrected in the revised version of the manuscript.
>
> **4. While majority of errors are in a range of [-100, 100], E1 seems to be in a range of [0, 100]. Should this be accounted for by rescaling it when calculating the composite score?**
>
> Great observation. This is due to the nature of E1 as a short-term forecasting error over a few time steps. All models outperform the trivial baseline for such short horizons, and as a result, negative scores (which would indicate worse-than-baseline performance) are not observed. We have added a note clarifying this in the manuscript.
>
> **5. Maybe this might be unfeasible to add to the metrics at this stage, but can the authors add some comments on the importance of also calculating computational resources used for training/inference and by numerical methods. One amortized metric is provided in the already cited McGreivy and Hakim paper.**
>
> This is an important point. As suggested, we have now incorporated reporting of computational cost based on the amortized metric defined in Equation (1) of McGreivy and Hakim (2023). We agree that reporting training/inference cost is vital for evaluating scientific ML methods, especially when assessing trade-offs between accuracy and efficiency. We have included a preliminary version in the appendix and plan to standardize and expand this reporting in future iterations of the benchmark.

---

> > ### Comment · Reviewer_qUAq · 2025-08-03
> >
> > After reading the authors’ rebuttal, I am satisfied that my main concerns have been addressed. The inclusion of SST dataset makes this a much stronger benchmark for PIML. I appreciate the inclusion of related benchmarks, concise table for errors and the amortized computational score in the appendix.
> >
> > The explanation for E1’s strictly positive range is reasonable, though I still encourage the authors to test whether future datasets might warrant rescaling the composite score.
> >
> > These edits are not yet reflected in the uploaded manuscript and supplementary file, and assuming they will be incorporated, I keep my score at 5.

---

> > ### Author Response · Authors · 2025-08-04
> >
> > We appreciate the reviewer's comments and careful consideration.
> >
> > Re: "edits are not yet reflected in the uploaded manuscript and supplementary file"
> >
> > Due to the updated NeurIPS's review process guidelines we were only allowed a 10'000 character, text-only rebuttal: "[...]
> >  - Because of technical complications, we need to stop supporting the global response with PDF. We increased the character limit per review from 6,000 to 10,000 to compensate for this.
> >  - Because of known concerns on identity leakage, we prohibit using any links in the rebuttal, including but not limited to anonymous or non-anonymous URL links, or updating your submitted github repository. [...]"
> > — Excerpt from email to authors.

---

> > > ### Comment · Reviewer_qUAq · 2025-08-04
> > >
> > > I just checked and I received that email as a reviewer too but missed it. Thank you for pointing that out. I recommend that the manuscript be accepted assuming the changes are incorporated in the camera ready version.

---

### Note · Authors · 2025-08-15

In this work we propose a Common Task Framework (CTF) for scientific machine learning (SciML). We set out to provide a challenging set of benchmarks to compare the rapidly growing collection of SciML models being developed. As an initial step, we provided the community with two canonical nonlinear systems: Kuramoto-Sivashinsky and Lorenz. We evaluate the performance of 12 models, in addition to two naïve baselines, on a diverse set of metrics that provide a holistic view of each model’s capabilities.

Our long-term vision is to take this initial step and grow the CTF to encompass a variety of scientific disciplines in order to objectively quantify which models perform well in specific domains for specific tasks. With the help of the reviewers, we have already grown the CTF by adding 6 foundation models and an additional high-resolution real-world dataset: daily sea-surface temperature (SST) observations from NASA.

We also plan on releasing a Kaggle competition to further promote engagement from the scientific community. Providing research groups the opportunity to improve their scores  by developing new models or improving existing ones is paramount to advancing the SciML field, similar to what has been done in computer vision.

We are extremely thankful for the opportunity to present our work to the broader community and are grateful to the reviewers for allowing us to strengthen our paper with their valuable feedback and continued engagement during the discussion period. We would also like to thank the ACs for their prompt responses during the discussion period.

---

### Decision · Program_Chairs · 2025-09-18

**Decision:**

Accept (poster)

**Comment:**

The paper introduces the Common Task Framework as a standardized evaluation method in the field of scientific machine learning. In total, twelve diverse evaluation metrics are proposed, and an extensive set of state-of-the-art methods is being evaluated. The paper proposes a novel approach borrowing from CV and NLP domains; a leaderboard based on hidden test data will be published on Kaggle, allowing for even more comprehensive evaluation of available methods. Initially, two datasets generated from chaotic dynamical systems were proposed; however, after the rebuttal phase authors added a real-life dataset based on NASA readings. The datasets proved to be very challenging for the available state-of-the-art methods; some of them did not even surpass a simple baseline. Moreover, during the rebuttal phase, several additional experiments were performed, significantly widening the scope of evaluation.

All of the reviewers agree that the paper should be accepted; therefore, I accept it for inclusion in the NeurIPS DB Track, provided that the final revision incorporates all additional experiments from the rebuttal phase.

===== FINAL UPDATE FROM DB Track PCs ====

The final decision for this paper has been taken by the program chairs after consultation with the SACs. All Senior Area Chairs have ranked papers according to the feedback from the AC during the review process. We decided to leave the original meta-review to reflect the opinion of the AC in light of the initial discussions with reviewers and SAC.